# Interpretable Sequence Classification Via Prototype Trajectory

## Abstract

We propose a novel interpretable recurrent neural network (RNN) model, called ProtoryNet, in which we introduce a new concept of prototype trajectories. Motivated by the prototype theory in modern linguistics, ProtoryNet makes a prediction by finding the most similar prototype for each sentence in a text sequence and feeding an RNN backbone with the proximity of each of the sentences to the prototypes. The RNN backbone then captures the temporal pattern of the prototypes, to which we refer as *prototype trajectories*. The prototype trajectories enable intuitive, fine-grained interpretation of how the model reached to the final prediction, resembling the process of how humans analyze paragraphs. Experiments conducted on multiple public data sets reveal that the proposed method not only is more interpretable but also is more accurate than the current state-of-the-art prototype-based method. Furthermore, we report a survey result indicating that human users find ProtoryNet more intuitive and easier to understand, compared to the other prototype-based methods.

## 1 Introduction

Recurrent neural networks (RNN) have been widely adopted in natural language processing. RNNs achieve the state-of-the-art performance by utilizing the contextual information in a "memory" mechanism modeled via hidden/cell states. Albeit the benefit, however, the memory mechanism obstructs the interpretation of model decisions: as hidden states are carried over time, various pieces of information get intertwined across time steps, making RNN models a "black box" inherently.

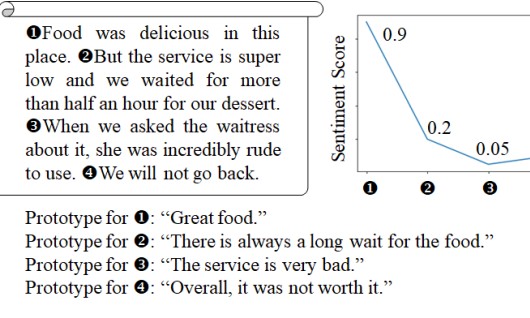

Prototype for ❶: "Great food."
Prototype for ❷: "There is always a long wait for the food."
Prototype for ❸: "The service is very bad."
Prototype for ❹: "Overall, it was not worth it."

Figure 1: Prototype trajectory-based explanation.

The black box nature of RNNs has motivated a body of research works aiming to achieve the *interpretability*. One approach is to leverage certain architecture design in the DNNs such as *Attention-based* methods. As will be discussed in Section 2, the attention-based approaches (Karpathy et al., 2015; Strobelt et al., 2017; Choi et al., 2016; Guo et al., 2018) visualize the RNN using the *attention mechanism*, which weighs the importance of each hidden state element. However, while a few of them could be quite illuminative, the attention weights are, in fact, not always intelligible. Rather, they often turn out to be a gibberish collection of numbers that does not possess much sensical interpretations. In fact, recent research has been considering attention weights as not explanations (Jain & Wallace, 2019). Furthermore, the analysis of attention weights requires a certain level of understanding of how RNNs work in theory. Hence, a novice user may find it difficult to understand and, thus, the broader use in real-world applications might not be so feasible.

The other is prototype-based approaches (Ming et al., 2019), which use prototypes to explain the decision more intuitively. The process is analogous to how, for example, human doctors and judges make decisions on a new case by referring to similar previous cases: for a given sequence, a prototype-based approach looks up a few representative examples, or *prototypes*, from the data set and deduces a decision. From the interpretability standpoint, such prototypes then provide intuitive clues and evidences of how the model has reached a conclusion in a form that even a layperson can understand.

However, the existing prototype-based methods find the prototypes at the whole-paragraph level, making it difficult to break down the analysis at the individual sentence level, *e.g.*, the connections and flows of individual sentences constituting a paragraph. Moreover, there may not be a suitable prototype when the length of a sequence is too large (*e.g.* a long paragraph), as longer sequences have greater degrees of freedom and, thus, harder to find a matching prototype, as evidenced later in Section 4.

Here, we advocate the idea that the sentence level prototyping (as opposed to the paragraph level prototyping in the previous literature) produce more desirable outcomes, namely better interpretability and higher prediction accuracy. We propose a novel architecture, called *ProtoryNet*, in which we introduce a new concept of the *prototype trajectory*. Given one or more paragraphs, ProtoryNet looks up the nearest prototype for each sentence and computes the proximity. The prototype proximity values are then fed into an RNN backbone, which then captures the latent patterns across sentences by means of the trajectory of prototypes. The prototype trajectory, therefore, illuminates the semantic structure of a text sequence and the logical flow therein, and, hence, provides highly intuitive, useful interpretation of how the model has predicted an output, as can be witnessed in Figure 1.

In fact, the prototype theory in modern linguistics supplies a strong justification for the proposed idea of prototype trajectory. In the prototype theory, linguists view "a sentence as the smallest linguistic unit that can be used to perform a complete action," (Alston, 1964) and analyze texts with individual sentences as building blocks. Linguists assume that the sentences of a category are distributed along a continuum: at one extreme of this continuum are sentences having a maximal number of common properties; while on the other extreme are sentences that have only one or very few of such properties (Panther & Köpcke, 2008). Here, the "ideal" sentence that possesses the maximally shared common properties can be considered as a *prototypical sentence*, or a *prototype* of the category. Thus, in some sense, this paper takes a meaningful first step towards mathematically formalizing the prototype theory in modern linguistics and its analysis methods by incorporating the above view in a computational framework and emulating how linguists analyze a text.

As such, ProtoryNet permits a fine-grained understanding of sequence data alongside an intuitive explanation of the dynamics of the subsequences. In addition, since the technical details are hidden in the prototypes, a non-technical user can comprehend the interpretation. However, when necessary, technical users, *i.e.*, the ones that are more knowledgeable about RNNs, can still look at the coefficients in RNN, similar to how the attention approaches visualize RNNs, as the proximity vectors feeding the RNN backbone are essentially one-hot encoded (*i.e.*, zero everywhere except the $k$-th position for prototype $k$), making it convenient to trace how coefficients are related to each prototype.

## 2 RELATED WORK

In addition to model-agnostic black-box explainers such as LIME (Ribeiro et al., 2016) and SHAP(Lundberg & Lee, 2017), various post hoc explanation methods have been proposed for DNN models, such as Integrated Gradients (Sundararajan et al., 2017), DeepLift (Shrikumar et al., 2017) and NeuroX (Dalvi et al., 2019). Specifically, to understand RNN models, Tsang et al. (2018) proposes a hierarchical explanations for neural networks to capture interactions and Jin et al. (2019) adapts the idea to text classification to quantify the importance of each word and phrase. For sentiment analysis, Murdoch et al. (2018) proposes contextual decomposition method for analyzing individual predictions made by standard LSTMs, and the method is able to reliably identify words and phrases of contrasting sentiment, and how they are combined to yield the LSTMs final prediction.

In addition to the external explanation methods, the prior efforts to bring interpretability to RNNs can be categorized as *attention-based* and *prototype-based* approaches. Bahdanau, Cho, and Bengio (Bahdanau et al., 2014) proposed an encoder-decoder type machine translation algorithm, in which they implemented an attention mechanism in the decoder. By the means of the alignment probabilities and the association energy, reflecting the importance of a given word in predicting a translated word, they let the attention mechanism to weigh which part of the source sentence the model needs to pay attention to. This not only improved the performance of the model by relieving the burden of the encoder having to compress all the information about the source sentence into a fixed-length vector, but also inherently visualized how the translation was conducted through the alignment matrix (*e.g.*, Figure 3 of Bahdanau et al. (2014)). Similarly, Rocktäschel et al. (2015) analyzed word-to-word attention weights for achieving insights into how a long short term memory (LSTM) classifier reasons about entailment. Zhang et al. (2017) proposed a language model to read and explore discriminative

image feature descriptions from reports to learn a direct mapping between lexical components and image pixels via attention. Similar strategies can be found in a number of other works (*e.g.* Ismail et al. (2019); Choi et al. (2016)) that employ the idea of the attention mechanism. However, the attention-based approaches are mostly intended for expert users. Many non-technical users in the real world, who lack basic knowledge of how RNNs work (or even neural networks in general), may find them difficult to understand.

Instead, prototype-based approaches argue that the intuitiveness of interpretation can be significantly enhanced by visualizing the reasoning process in terms of prototypes. In fact, prototype-based reasoning has a long history as a fundamental interpretability mechanism in traditional models (Cupello & Mishelevich, 1988; Fikes & Kehler, 1985; Kim et al., 2014). More recently, the idea of prototype-based interpretation attracted several authors in neural network research. For instance, Chen et al. (2019) incorporated the concept of prototypes to convolutional neural networks. A prototype layer was added after convolutional layers to compare the convolution responses at different locations with prototypes. From this, users can understand, for example, a bird is classified as a 'red-bellied woodpecker' because it has the typical prominent red tint at the belly and the top of its head as well as the black and white stripes on its wings.

More closely related to the present work, ProSeNet (Ming et al., 2019) added a prototype layer in RNN. ProSeNet computed the similarities between an input sequence (usually a short prose) and prototypes and produced the final prediction as a linear combination of the similarities. Despite the built-in interpretability of ProSeNet, however, an issue might arise when the sequence was too long. The original paper (Ming et al., 2019) validates ProSeNet only on paragraphs shorter than 25 words. However, it is easily fathomable that ProSeNet may fail to assimilate a long paragraph data, due to large degrees of freedom that complicate matching of a prototypical example. This may render some practical concerns. For instance, in sentiment classification, even if a paragraph is classified as "negative," it could consist of several twists of sentiments along sentences (*e.g.*, sarcastic use of positive proses). With an increased length, such kinds of twists can get harder to be represented with a prototype, thus making the interpretation difficult and the explanation less credible. This claim is further supported by findings in modern linguistics, which suggests that sentences, instead of paragraphs, should be regarded as the basic elements for text analysis (Panther & Köpcke, 2008).

Nonetheless, we share our view with ProSeNet in that the prototype based method is worth paying attention to. The benefit of prototype-based reasoning resides in the fact that it hides technical details by encapsulating them with prototypical examples while being still tractable numerically when desired. Hence, novice users can understand how the reasoning was achieved in RNNs so long as they can comprehend the prototypes, lowering the barrier for those numerous non-technical users who may use RNN-based applications in the real world. On the other hand, numerical weights assigned to prototypes alongside their association with the "nuts and bolts" of RNNs still allow experts to perform in-depth analyses of how a model has drawn a prediction.

## 3 METHOD

### 3.1 THE PROTORYNET ARCHITECTURE

Suppose a data set $\mathcal{D} = \left\{ (\mathbf{X}^{(i)}, \mathbf{y}^{(i)}) : i = 1, \dots, N \right\}$ of size $N$, comprised of text sequences $\mathbf{X}^{(i)}$ and the corresponding labels $\mathbf{y}^{(i)}$. Here, note that the superscript $(i)$ may be dropped for notational convenience hereinafter, unless necessary. Each instance $\mathbf{X}$ can be understood as a sequence of sentences $\mathbf{x}_t \in \mathbb{R}^V$ at $t$-th position, yielding the representation $\mathbf{X} = (\mathbf{x}_t)_{t=1}^{T}$, where $V$ is the size of vocabulary and $T := |\mathbf{X}|$ is the number of sentences in the sequence $\mathbf{X}$. $\mathbf{y} \in \mathbb{R}^C$ is a multi-hot encoded vector representing the class labels associated with the sequence $\mathbf{X}$, *i.e.*, the $c$-th element $y_c$ of $\mathbf{y}$ equals 1 if the label $c$ is associated with $\mathbf{X}$ or 0 otherwise. $C$ is the total number of classes.

ProtoryNet interfaces with text data via a sentence encoder (Figure 2a) modeled as a mapping $\mathbf{r} : \mathbb{R}^V \to \mathbb{R}^J$, where $J$ is the dimension of sentence encoding specified by the user. That is, the encoder takes each sentence $\mathbf{x}_t \in \mathbf{X}$ and produces a sentence embedding $\mathbf{e}_t = \mathbf{r}(\mathbf{x}_t)$. In this paper, the development of the encoder $\mathbf{r}$ is beyond the scope of this paper and, hence, we employ a pre-trained BERT-based sentence encoder with mean-tokens pooling (Reimers & Gurevych, 2019), where $J = 768$ by default.

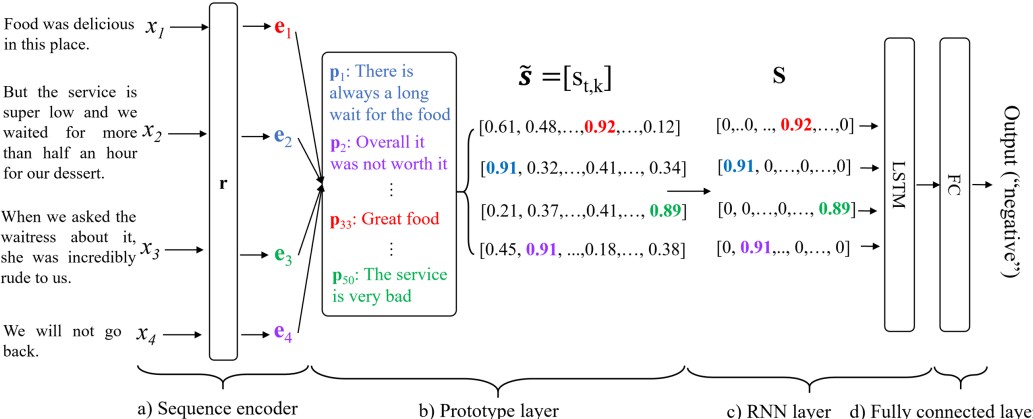

Figure 2: The architecture of ProtoryNet.

The sentence embeddings $\mathbf{e}_t$ are fed into the *prototype layer* (Figure 2b), in which a set of trainable prototypes $\mathcal{P} = \left\{ \mathbf{p}_k \in \mathbb{R}^J : k = 1, \ldots, K \right\}$ are compared with $\mathbf{e}_t$, where $K := |\mathcal{P}|$ is the number of prototypes specified by the user. Then, given a distance metric $d : \mathbb{R}^J \to \mathbb{R}^+$, the proximity $s_{t,k}$ of the sentence embedding $\mathbf{e}_t$ to a given prototype $\mathbf{p}_k$ is measured as $s_{t,k} := s(\mathbf{e}_t, \mathbf{p}_k) = \exp\left( -\frac{d(\mathbf{e}_t, \mathbf{p}_k)}{\psi^2} \right)$, where $\psi \in \mathbb{R}$ is a user-specified constant. Between two popular choices for the distance metric $d$, namely the cosine distance and the Euclidean distance, we find that there is no significant difference between the two. Hence, we use the Euclidean distance in our experiments for convenience.

Note here that the intermediate throughput of the prototype layer is the similarity matrix $\tilde{\mathbf{S}} = [s_{t,k}]$ of the size $T \times K$, associating the $t$-th sentence with the $k$-th prototype. The rows of the similarity matrix $\tilde{\mathbf{S}}$ then constitute the input to the LSTM backbone at time step $t$ (Figure 2c), which then finally produces an output prediction. Here, the *sparsity transformation* is performed on the similarity matrix $\tilde{\mathbf{S}}$ before feeding it into the LSTM backbone by setting each row to zero except for the position where $s_{t,k}$ is the maximum. That is, each row of the transformed similarity matrix $\mathbf{S}$ would be of the same topology as the one-hot encoded vector, whose elements equal $s_{t,k^*}$ at $k^* := \arg\max_k s_{t,k}$ and 0 otherwise. The sparsity transformation of $\tilde{\mathbf{S}}$ to $\mathbf{S}$ enhances the interpretability of the architecture, by enforcing each sentence be matched with no more than a single prototype and, thus, disentangling the information. This is accomplished only at a small cost of accuracy, as observed from an ablation study in Appendix.

**Motivating Example**   The text data in Figure 2 exemplifies the use of ProtoryNet for sentiment analysis (text classification). In this example, the task is to predict whether the review of a restaurant is positive or not. The input text data $\mathbf{X}$ is comprised of $T = 4$ sentences, in this particular case, and the label $\mathbf{y}$ is the binary sentiment label of the review, either "positive" ($[1, 0]$) or "negative" ($[0, 1]$). To this, ProtoryNet converts the text data into sentence embeddings, each of which are then matched with the closest prototype. Observe, in the figure, that the prototypes that ProtoryNet produced are, indeed, morphosyntactically equivalent to the corresponding input sentences, well-exemplifying them semantically. The one-hot-like similarity vectors between the sentences and the prototypes are then fed into the LSTM backbone, which captures the patterns and trends in the trajectory of prototypes and, finally, predicts the final sentiment label, which, in this case, is "negative."

## 3.2   OBJECTIVE FUNCTIONS

The training objectives of ProtoryNet entail four different terms aiming to achieve both the prediction accuracy and the interpretability. Below are the details of their definitions.

**Accuracy**   The *accuracy loss* is defined as the square loss between the predicted value and the ground truth label, promoting the model to make accurate predictions:

$$\mathcal{L}_{\mathrm{acc}}(\mathcal{D}) := \frac{1}{N} \sum_{i=1}^{N} \left\| \mathbf{y}^{(i)} - \hat{\mathbf{y}}^{(i)} \right\|^2. \tag{1}$$

**Diversity** To ensure diverse and non-overlapping prototypes, we define the *diversity loss* term added to enforce the minimum mutual distance $\delta$ among the prototypes:

$$d_{\min} := \min_{k_1, k_2} d(\mathbf{p}_{k_1}, \mathbf{p}_{k_2}), \tag{2}$$

$$\mathcal{L}_{\mathrm{div}}(\mathcal{D}) := \sigma\left(\eta(\delta - d_{\min})\right), \tag{3}$$

where $\sigma(\cdot)$ is the sigmoid function and $\eta$ is a smoothing constant, which we set $\eta = 1$ empirically. The constant $\delta \in \mathbb{R}_*^+$ is a positive real number defined by the user, to enforce the minimum separation among prototypes. Hence, when the distances among the prototypes do not meet the minimum separation requirement *i.e.*, $d_{\min} < \delta$, the $\eta(\delta - d_{\min})$ term will have some positive value, making the diversity loss term $\mathcal{L}_{\mathrm{div}}$ active; on the other hand, when the minimum separation requirement is met and thus, $d_{\min} > \delta$, then the sigmoid function will pull the loss term to zero. Note that a smaller $\eta$ will make such a transition by the sigmoid function smoother.

**Prototypicality** The accuracy and the diversity terms alone, it is observable a prominent tendency of prototypes diverging away from the sentences during training. Such a behavior introduces overfitting, in which prototypes become less generalizable, as the prototypes lose their representativity of a category. Hence, we introduce the *prototypicality* loss, which promotes each sentence in the database to have a representative prototype close to it:

$$\mathcal{L}_{\mathrm{proto}} = \frac{1}{S} \sum_{\mathbf{X} \in \mathcal{D}} \sum_{\mathbf{x}_t \in \mathbf{X}} \min_k d(\mathbf{r}(\mathbf{x}_t), \mathbf{p}_k), \tag{4}$$

where $S$ is the total number of sentences in the data set.

**Final loss** The final loss function combines the above loss terms:

$$\mathcal{L} = \mathcal{L}_{\mathrm{acc}} + \alpha \mathcal{L}_{\mathrm{div}} + \beta \mathcal{L}_{\mathrm{proto}} \tag{5}$$

Empirically, coefficients values of $\alpha = 0.1$ and $\beta = 1e^{-4}$ are used in this paper.

**Remarks on Prototype Interpretability** The diversity and prototypicality terms are designed for improving the interpretability. Here, to achieve good explanations, prototypes need to be different from each other to avoid redundancy, thus the diversity term. In addition, each input sentence needs to be mapped to a prototype that is similar enough to make the explanation convincing, thus the prototypicality term. These two loss terms can be considered regularization terms to serve interpretability purposes. Similar loss terms have been introduced in other prototype based DNN models (Ming et al., 2019; Chen et al., 2019). We will later show in experiments that these two terms do not hurt the predictive performance. This can be explained by the recent research on Rashomon Set (Semenova et al., 2019; Rudin, 2019), that there exist many models with very similar performance, so one can add customized constraints to the model to achieve additional benefits, such as interpretability.

### 3.3 TRAINING

For the training of ProtoryNet, the adaptive moment estimation (ADAM) optimizer (Kingma & Ba, 2014) was employed. The learning rate was set to be $1e^{-4}$ and the exponential decay rates for the first and the second moment estimates were 0.9 and 0.999, respectively. Below are further details used for generating the results in this paper.

**Differentiability** Training of ProtoryNet requires the computation of the index where the similarity matrix $\tilde{\mathbf{S}}$ is maximum for the sparsity transformation in the prototype layer. This operation, unfortunately, is not differentiable and may lead to an unexpected training behavior during auto-differentiation in deep learning packages. We get around this issue by the following approximation technique. Suppose the similarity matrix $\tilde{\mathbf{S}} = [\tilde{\mathbf{s}}_1, \ldots, \tilde{\mathbf{s}}_T]$ where $\tilde{\mathbf{s}}_t \in \mathbb{R}^K$ is a column vector whose elements indicate how similar the $t$-th sentence is to each of the prototypes. If we let Softmax$(\cdot)$ to denote the softmax function, then for some large constant $\gamma$,

$$\mathbf{\Gamma} = [\mathrm{Softmax}(\gamma \cdot \mathbf{s}_1), ..., \mathrm{Softmax}(\gamma \cdot \mathbf{s}_T)] \tag{6}$$

approximates the selection matrix whose element equals to 1 at the position corresponding to where $\mathbf{s}_t$ is the maximum for each column $t$ and 0 elsewhere. Here, we find $\gamma \geq 1e^6$ gives a reasonable

approximation empirically. With the selection matrix, the sparsity transformation can be approximated as follows without explicitly computing the maximum:

$$\mathbf{S} \approx \mathbf{\Gamma} \odot \tilde{\mathbf{S}} \tag{7}$$

where $\odot$ is the Hadamard product. Note that the softmax function is differentiable and, thus, is $\mathbf{S}$.

**Prototype Initialization** The training of ProtoryNet can benefit from the initialization method described below. We first embed all sentences separately in the training data set. Then, in the embedding space, all sentences in the data set are clustered using the $k$-medoids clustering algorithm to categorize sentences by their semantic meaning. The medoids obtained from the $k$-medoids algorithm can be considered as the representative examples of each cluster and, hence, plausible candidates for prototypes. Thus, for the training of ProtoryNet, we use these medoids to initialize prototypes, which in turn accelerates the convergence.

**Prototype Projection** It should be noted that the numerical solutions for the prototypes are found in the embedding space. These numerical solutions are not automatically intelligible to human users and need to be deciphered. To this end, we project the prototypes to the closest sentence in the embedding space every 10 epochs during the training process, similar to the technique proposed in Ming et al. (2019); Chen et al. (2019):

$$\mathbf{p}_k = \operatorname*{arg\,min}_{\mathbf{x}_t \in X^{(i)}, \forall X^{(i)} \in \mathcal{D}} d(\mathbf{r}(\mathbf{x}_t), \mathbf{p}_k), \qquad k \in [1, K] \tag{8}$$

**Sentiment Scores for Prototypes** Once the training is done, ProtoryNet returns a set of $K$ prototypes. We then feed them back into the trained ProtoryNet one at a time, to evaluate the sentiment score of each individual prototype. These sentiment scores will later be used to provide quantitative visualizations of how the tones and sentiments change within a text data.

## 4 EXPERIMENTS

In this section, we evaluate our model on five different data sets. A vanilla LSTM method, a state-of-the-art black-box model (DistilBERT Sanh et al. (2019)), and a state-of-the-art prototype-based interpretable model (ProSeNet Ming et al. (2019)) are compared against our method. We also compare with a classic non-neural baseline bag-of-words, which provide explanations at word level.

| Data set | DistilBERT | Vanilla LSTM | ProSeNet | **ProtoryNet** | ProtoryNet (avg) | Bag-of-words |
|---|---|---|---|---|---|---|
| IMDB | .923 ($\pm$.001) | **.873** ($\pm$.007) | .835 ($\pm$.008) | .849 ($\pm$.002) | .801 ($\pm$.002) | .870 ($\pm$.004) |
| Amazon Reviews | .941 ($\pm$.004) | .871 ($\pm$.004) | .840 ($\pm$.011) | **.882** ($\pm$.004) | .860 ($\pm$.005) | .869 ($\pm$.005) |
| Yelp Reviews | .957 ($\pm$.003) | .897 ($\pm$.004) | .868 ($\pm$.008) | .872 ($\pm$.002) | .859 ($\pm$.004) | **.902** ($\pm$.010) |
| Rotten Tomatoes | .841 ($\pm$.001) | .751 ($\pm$.004) | .748 ($\pm$.006) | **.762** ($\pm$.007) | .656 ($\pm$.010) | .749 ($\pm$.005) |
| Hotel Reviews | .976 ($\pm$.006) | .947 ($\pm$.004) | .909 ($\pm$.007) | **.949** ($\pm$.004) | .856 ($\pm$.006) | .944 ($\pm$.008) |

Table 1: Performance of ProtoryNet in comparison with other benchmark models. The mean accuracy and the standard deviation from a 5-fold cross validation are reported in each case. Boldface denotes the best performing model, excluding the black box model, DistilBERT, which outperformed the others in all cases. Note each data set had balanced labels.

### 4.1 PREDICTION ACCURACY

Reported in Table 1 are the means and the standard deviations of accuracy evaluated from 5-fold cross validations. It is clear that the state-of-the-art black box model (DistilBERT) outperforms the interpretable models (ProSeNet and ProtoryNet) in all data sets used. Note, however, that DistilBERT was pre-trained on a massive corpus of text data before being transferred to each specific data set. Hence, the performance metrics of DistilBERT should only be used for sanity check. Compared against the vanilla LSTM, the prediction performance of the interpretable models were on par. ProtoryNet is slightly better than Bag-of-words on average.

Between ProSeNet and ProtoryNet, ProtoryNet outperformed ProSeNet for all five cases overall. In particular, the performance difference was clearer when long text data were analyzed. In Table 2, we split each data set into *short* and *long* samples—paragraphs that were less than 25 words were classified as short samples, following the criterion used in the ProSeNet paper Ming et al. (2019). As

| Data set | % of short reviews | ProSeNet (Ming et al., 2019) | | ProtoryNet (ours) | |
|---|---|---|---|---|---|
| | | Short | Long | Short | Long |
| IMDB | 0.09 | **0.883** (±0.126) | 0.835 (±0.008) | 0.865 (±0.127) | **0.849** (±0.009) |
| Amazon | 5.08 | 0.868 (±0.014) | 0.833 (±0.009) | **0.879** (±0.034) | **0.882** (±0.004) |
| Yelp | 9.22 | **0.879** (±0.018) | 0.859 (±0.016) | 0.794 (±0.005) | **0.879** (±0.002) |
| Rotten Tomatoes | 61.37 | 0.754 (±0.009) | 0.721 (±0.015) | **0.774** (±0.006) | **0.738** (±0.015) |
| Hotel reviews | 2.22 | 1.000 (±0.000) | 0.904 (±0.006) | 1.000 (±0.000) | **0.948** (±0.024) |

Table 2: Comparison between ProSeNet and ProtoryNet on text data of different lengths. In the table are the means and the standard deviations of accuracy over 5-fold cross validation. The two methods perform similarly on short reviews ($\leq$ 25 words), while ProtoryNet performs better on long reviews in all data sets. The second column represents the proportion of short reviews in each data set.

shown in the table, ProSeNet and ProtoryNet were on par when short proses were analyzed, while ProtoryNet was always better than ProSeNet when long paragraphs were concerned. In fact, this can be a substantial advantage of ProtoryNet as instances with less than 25 words are quite rare in real-world data sets. Note, in Table 2, more than 90% of instances are more than 25 words, with an exception of the Rotten Tomatoes data set.

In addition, we investigated how the number of prototypes, $K$, influences the performance of ProtoryNet. In Figure 3, the performance of ProtoryNet on the Amazon Review data set is plotted with respect to different values of $K$. Other hyperparameters were controlled to be the same and the performance measures (accuracy) were averaged over 5-fold cross validation experiments (the whiskers in the figure represent the standard deviation). Compared to the ProSeNet baseline, ProtoryNet reaches high accuracy much quicker with only a few prototypes and shows a steady increase of performance afterwards. The same analyses on the other data sets are available in Appendix.

Figure 3: Effect of $K$ on accuracy.

Furthermore, we conducted an ablation study on the sparsity transformation. We measured the change in prediction accuracy when the sparsity transformation step had been removed and the dense similarity matrix $\tilde{\mathbf{S}}$ had been used directly. The result revealed that there was only a small drop of accuracy (approx. 1%) caused by the sparsity transformation, while the benefit of interpretability was huge (see Appendix for the detail).

## 4.2 PROTOTYPE TRAJECTORIES

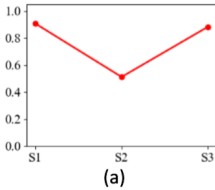
(a)

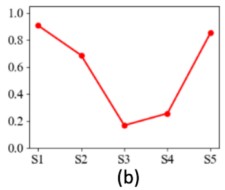
(b)

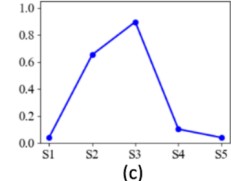
(c)

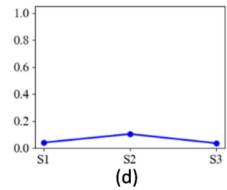
(d)

Figure 4: Prototype trajectories of two positive sentiment examples (a, b) and two negative samples (c, d). The corresponding prototypes are available in Appendix.

Prototypes and their trajectory play a critical role in ProtoryNet. Here, we show four examples of prototype trajectories in Figure 4, two for positive sentiment data and two for negative ones curated from the Yelp Review data. Observe that the trajectories can be drastically different even for the

same sentiment class. For example, review (c) starts off with a negative sentence and changes the tone in the middle to positive and then ends with a negative sentence while (d) maintains the negative tone from the beginning to the end. As such, interpretation of ProtoryNet can be more fine-grained, generating a deeper insights to users.

### 4.3 THE EFFECT OF DIVERSITY AND PROTOPYPICALITY TERMS

The diversity and prototypicality terms are designed to improve interpretability since it forces each sentence to be mapped to a prototype that is close enough to it and prototypes to be sufficiently different from each other. Therefore, in this section, we examine whether the two terms serve such purposes, and meanwhile, whether these two terms hurt the predictive performance.

**Effect on Interpretability** We plot two types of distances for hotel dataset. First, since the explanations are based on prototypes, we need the prototypes to be similar enough to the input sentences mapped to them in order to make the explanation convincing. Thus we compute the Euclidean distances between each sentence and the prototypes they are mapped to. Second, we want prototypes to be at

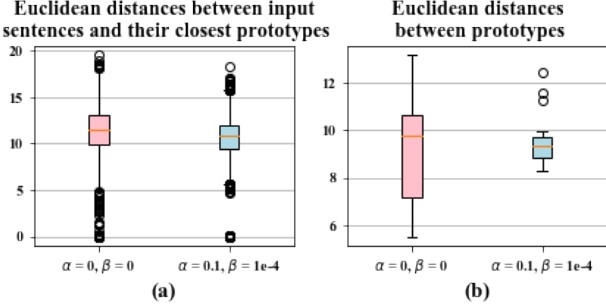

Figure 5: The effect of $\alpha, \beta$ on interpretability

least $d_{\min}$ (in this experiment we choose $d_{\min} = 8$) away from each other, since if prototypes are too similar, then similar sentences may be mapped to different prototypes, causing confusion to users. Therefore, we compute the Euclidean distances between each pair of prototypes. Figure 5 shows, when setting $\alpha, \beta$ non-zero, the distances between sentences and their prototypes are closer and prototypes are sufficiently away from each other, compared to when $\alpha, \beta = 0$, proving the effect of the diversity and prototypicality terms on interpretability.

**Effect on Accuracy** We performed a sensitivity analysis to understand the effect of the two terms on predictive performance. We trained ProtoryNet on the Amazon dataset with different combinations of $\alpha = 0, 1e^{-3}, 1e^{-2}, 1e^{-1}, 1$, and $\beta = 0, 1e^{-5}, 1e^{-4}, 1e^{-3}, 1e^{-2}$. As seen in Figure 6, our experiment revealed that the performance of ProtoryNet is not so sensitive to the selection of the parameters $\alpha$ and $\beta$. Between $\alpha$ and $\beta$, ProtoryNet was more sensitive to $\beta$, which may reveal the trade-off between the different loss terms. The results also illustrate how different loss terms impact the overall performance - when $\alpha = 0$ or $\beta = 0$. The best performance was achieved when $\alpha$ and $\beta$ are

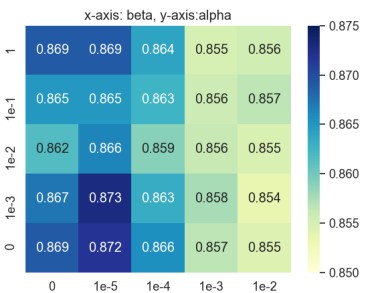

Figure 6: Sensitivity analysis.

set to small values instead of 0. A possible explanation would be that, having some constraints on the prototypes' diversity ($\mathcal{L}_{\text{div}}$) and their representativeness ($\mathcal{L}_{\text{proto}}$) prevents overfitting as these terms "regulate" prototypes. Sensitivity analysis for other datasets are included in the Appendix.

### 4.4 USER EVALUATION

The interpretability of ProtoryNet was further validated via a survey conducted on 111 individuals, among which 42 identified themselves as non-technical users. Subjects were recruited through two different channels. Individuals from the authors' home institution holding a master's degree or above having advanced knowledge of RNNs have been recruited as technical users. Non-technical users were recruited from Amazon Mechanical Turk. The summary statistics of the subjects as well as the survey design are disclosed in Appendix.

We first evaluated the interpretability of the explanation by testing whether the model-selected prototypes were indeed representative of the input text to the human users. We asked the users to choose the most appropriate prototype for a given sentence out of four options presented to them, one of which was the actual prototype matched by the model, other two were randomly selected from the rest of the prototypes, and the other was "None of the above." We created 10 such questions by sampling reviews from the Yelp Review data set, each for ProtoryNet and ProSeNet. As reported in

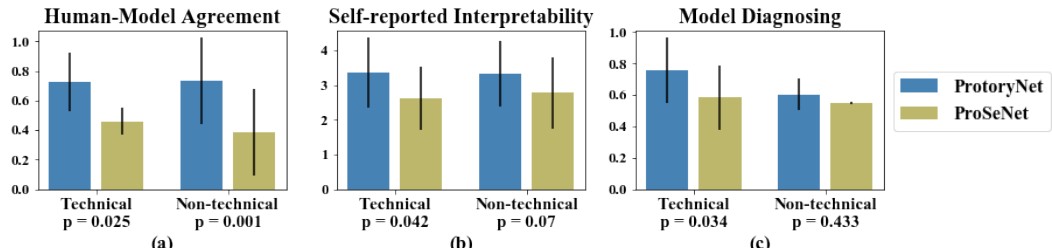

Figure 7: Interpretability of ProtoryNet assessed by human users. The p-values are evaluated for comparing the responses for ProtoryNet and ProSeNet on technical users and non-technical users, respectively.

Figure 7a, ProtoryNet showed a more significant agreement between the model-selected prototype and the prototype that the human users found the most appropriate. For both technical users and non-technical users, ProtoryNet was significantly better than ProSeNet, as was validated by the t-test. The difference between technical users and non-technical users was insignificant, suggesting that non-technical users can comprehend prototypes equally well as technical users.

The survey also included self-report questions to assess how easy it was for them to select a prototype in a score ranging between 1 (very difficult) and 5 (very easy). As reported in Figure 7b, subjects found that ProtoryNet was easier to interpret in general, and the improvement in interpretaiblity was more significant for technical users. Again, the difference between technical users and non-technical users was insignificant.

Finally, we measured how easily the users can learn to interpret the results of ProtoryNet. For this, each subject was randomly assigned to either ProtoryNet or ProSeNet and trained on how the model that they are assigned to makes predictions. Then, their proficiency was measured by showing them three examples on which the model had made an incorrect prediction and asking them to diagnose the problem by pointing out an inappropriately matched prototype. The problematic prototype (*i.e.*, the "correct answer" for the survey question) was determined via a discussion among the authors, which later turned out to be aligned with the consensus in the survey responses as well. As in Figure 7c, the both subject groups were more accurate at diagnosing ProtoryNet in general. An explanation to this should be that ProtoryNet uses shorter prototypes than ProSeNet and, thus, is easier to comprehend. We notice that the while technical users find ProtoryNet easier to debug, such difference was not significant for non-technical users. In fact, there was no significant difference between technical users and non-technical users when they use ProSeNet since it was almost equally difficult to these two groups of users.

## 5 CONCLUSION

In this paper, we introduced a novel idea of prototype trajectory in RNNs. Our model, ProtoryNet, utilizes the prototype trajectories to map a text data into a sequence of prototypical sentences, illuminating the underlying dynamics of semantics within the text data. We showed that ProtoryNet could achieve a predictive performance higher than the current state-of-the-art. In addition, sentence level prototype trajectory allowed fine-grained analysis of paragraphs. Moreover, the survey result suggested that ProtoryNet provided more intuitive prototypes than the state-of-the-art method and that the novice users were able to interpret ProtoryNet equally well as the expert users.

Our immediate future work would be to apply ProtoryNet to other types of sequence data, such as medical data, longitudinal data, etc. In a more theoretical context, it would be interesting to mathematically formalize some of the well-established requirements to be a prototype in the linguistics literature. For example, Panther and Köpcke (Panther & Köpcke, 2008) assert several conditions that a prototype must possess—a prototypical sentence is an affirmative declarative sentence; the subject is in the nominative case; the verb in a prototype is in the active voice and in the indicative mood; to list a few. Albeit non-trivial, the mathematical translation of such conditions should bring more interpretability and, perhaps, a better performance of ProtoryNet.

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

## 6 APPENDIX

### 6.1 REPRODUCIBILITY

#### 6.1.1 DATA SETS

**IMDB Movie Reviews** The IMDB Movie Reivews data set is a standard benchmark data set for binary sentiment classification and is available at `https://ai.stanford.edu/~amaas/data/sentiment/`. The data set is comprised of 25,000 movie reviews, divided equally into 12,500 positive and 12,500 negative reviews. The total of 332,541 sentences are found in the data set. All of those reviews samples were included in our experiments.

**Amazon Product Reviews** The Amazon Product Review data set is publicly available on Kaggle: `https://www.kaggle.com/bittlingmayer/amazonreviews`. In our experiment, we took random samples of a similar size to the IMDB data set—30,001 reviews were randomly selected from the original Amazon data set, among which 15,429 were positive and 14,572 were negative. The total of 192,709 sentences were found.

**Yelp Reviews** The Yelp Reviews data set was obtained from `http://goo.gl/JyCnZq`. The data set is comprised of 555,000 Yelp review samples and their corresponding labels. The authors of the data set has binarized the sentiment scores by assuming 1 and 2 stars as a negative sentiment and 3 and 4 stars as a positive sentiment. The total of 30,000 review samples were extracted via random sampling, in which the number of positive reviews were 14,389 and the number of negative reviews were 15,611.

**Rotten Tomatoes** The Rotten Tomatoes Movie Review data set is a corpus of movie reviews used for sentiment analysis and is available at `https://github.com/nicolas-gervais/rotten-tomatoes-dataset`. The total of 30,000 reviews were randomly selected, in which the proportion between the positive reviews and the negative reviews was exactly 1:1.

**Hotel Reviews** The Hotel Reviews data set is comprised of 20,000 review samples evaluating 1,000 hotels and is available on Kaggle: `https://www.kaggle.com/datafiniti/hotel-reviews`. In this paper, we assumed a positive sentiment for reviews of 4 and 5 star ratings and a negative sentiment for reviews of 1 and 2 stars. Reviews with 3 stars were ignored. This assignment yields 17,746 positive reviews and 2,254 negative ones. To balance out the data set, we randomly picked 2,254 positive reviews to make them equal, making the total of 4,508 reviews used in our experiments.

### 6.1.2 MODELS

**Vanilla LSTM** We used 300-dimensional GloVe word embeddings Pennington et al. (2014) to encode words in sentences. An LSTM model with 2 hidden layers of size 128 each was used. The final prediction was made by a fully connected layer of size 256. A dropout layer of the rate 0.5 was used immediately before the fully connected layer.

**DistilBERT** DistilBERT Sanh et al. (2019) is considered as a light-weight version of the state-of-the-art BERT model with smaller, faster, and less expensive deployment time and resources. In our experiments, a pre-trained DistilBERT model was transferred to each target data set.

**ProSeNet** ProSeNet Ming et al. (2019) is a state-of-the-art prototype-based interpretable RNN. For the implementation of ProSeNet, we used an LSTM layer with 2 hidden layers of size 128 and the dropout rate 0.5 for the sequence encoder. This is the same configuration as the ProtoryNet's RNN layer. We fixed $K = 200$ for all experiments for both ProSeNet and ProtoryNet, with an exception of Figure 3.

For fair comparison, we used the fixed constant $K = 200$ for both ProtoryNet and ProSeNet. In addition, the LSTM layer in ProtoryNet was implemented to have the same architecture as the baseline methods to eliminate the bias.

We used TensorFlow v1.15[1] to implement ProtoryNet and the other benchmark models including standard LSTM and ProSeNet. For DistilBERT, we used an implementation that was available in the Hugging Face Transformers Library (`https://github.com/huggingface/transformers`), which was implemented in PyTorch and TensorFlow 2.0.

For pre-processing, the period ('.'), the question mark ('?'), and the exclamation mark ('!') were used as delimiters to define the boundary between sentences. All words were then converted to the lowercase and punctuations were removed using the definition in `string.punctuation` constant in Python 3.5. In all experiments, we used pre-trained BERT-based language model with mean-tokens pooling Reimers & Gurevych (2019) to convert the raw sentence data to sentence embeddings.

### 6.2 ABLATION STUDY

### 6.2.1 SPARSITY TRANSFORMATION

In this paper, the sparsity transformation from $\tilde{\mathbf{S}}$ to $\mathbf{S}$ was used to enhance the interpretability of the model. However, the effect of the sparsity transformation on the model performance has not been investigated in the main text. Hence, we compare ProtoryNet's performance with and without the sparsity transformation.

As reported in Table 3, ProtoryNet without the sparsity transformation exhibits a better performance than when the sparsity transformation was used. Although the margin was narrow, such a trend was statistically significant. This should be an inevitable cost paid in exchange of the enhanced interpretability. A future investigation may include representing a sentence with more than a single prototype, to balance between the interpretability and the increased accuracy.

---

[1]https://www.tensorflow.org/

| Data set | Dense | Sparse |
|---|---|---|
| IMDB | **0.861** ($\pm$0.002) | 0.849 ($\pm$0.002) |
| Amazon Reviews | **0.893** ($\pm$0.015) | 0.882 ($\pm$0.004) |
| Yelp Reviews | **0.885** ($\pm$0.002) | 0.872 ($\pm$0.002) |
| Rotten Tomatoes | **0.796** ($\pm$0.003) | 0.762 ($\pm$0.003) |
| Hotel Reviews | 0.944 ($\pm$0.003) | **0.949** ($\pm$0.004) |

Table 3: Performance comparison between non-sparse $\tilde{\mathbf{S}}$ and sparse $\mathbf{S}$ as the input to the LSTM layer. The mean accuracy and the standard deviation from 5-fold cross validation are reported for each case. Boldface was used to indicate the one with a better performance.

### 6.2.2 DISTANCE METRICS

One of the design decisions we made for ProtoryNet architecture was between the cosine distance and the Euclidean distance metrics. However,as reported in Table 4, a comparative study revealed that there was no significant difference in performance.

| Data set | Cosine | Euclidean |
|---|---|---|
| IMDB | **0.879** ($\pm$0.004) | 0.849 ($\pm$0.002) |
| Amazon Reviews | 0.881 ($\pm$0.003) | **0.882** ($\pm$0.004) |
| Yelp Reviews | **0.879** ($\pm$0.005) | 0.872 ($\pm$0.002) |
| Rotten Tomatoes | 0.760 ($\pm$0.004) | **0.762** ($\pm$0.003) |
| Hotel Reviews | 0.946 ($\pm$0.008) | **0.949** ($\pm$0.004) |

Table 4: Performance comparison between the cosine distance and the Euclidean distance. The mean accuracy and the standard deviation from 5-fold cross validation are reported for each case. Boldface was used to indicate the one with a better performance.

### 6.2.3 INFLUENCE OF THE NUMBER OF PROTOTYPES ON ACCURACY

Here, we provide additional results to show how the number of prototypes $K$ impacts the model performance.

### 6.3 SENSITIVITY ANALYSIS

We provide sensitivity analysis on all datasets except the one in the main paper. The sensitivity analysis studies how the predictive performance varies with different values of $\alpha$ and $\beta$

### 6.3.1 LIST OF PROTOTYPES IN SECTION 4.2

Figure 10 shows the prototypes and their sentiment scores used in the main text Section 4.2 and the main text Figure 4. Note that several sentences could be mapped to one prototype.

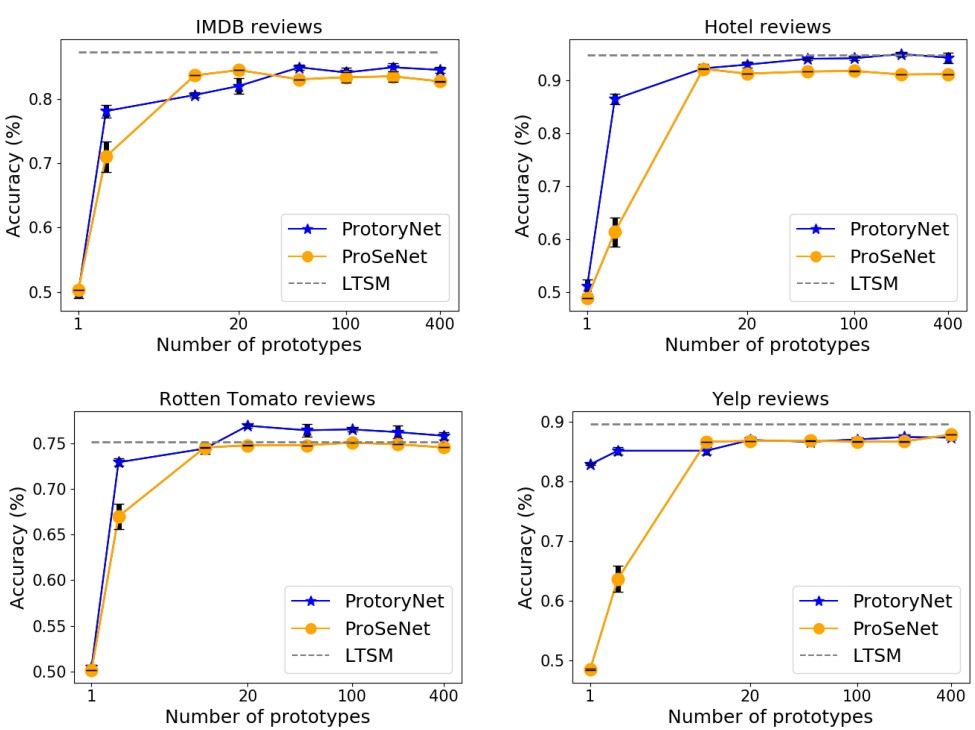

Figure 8: Effect of $K$ on the model accuracy.

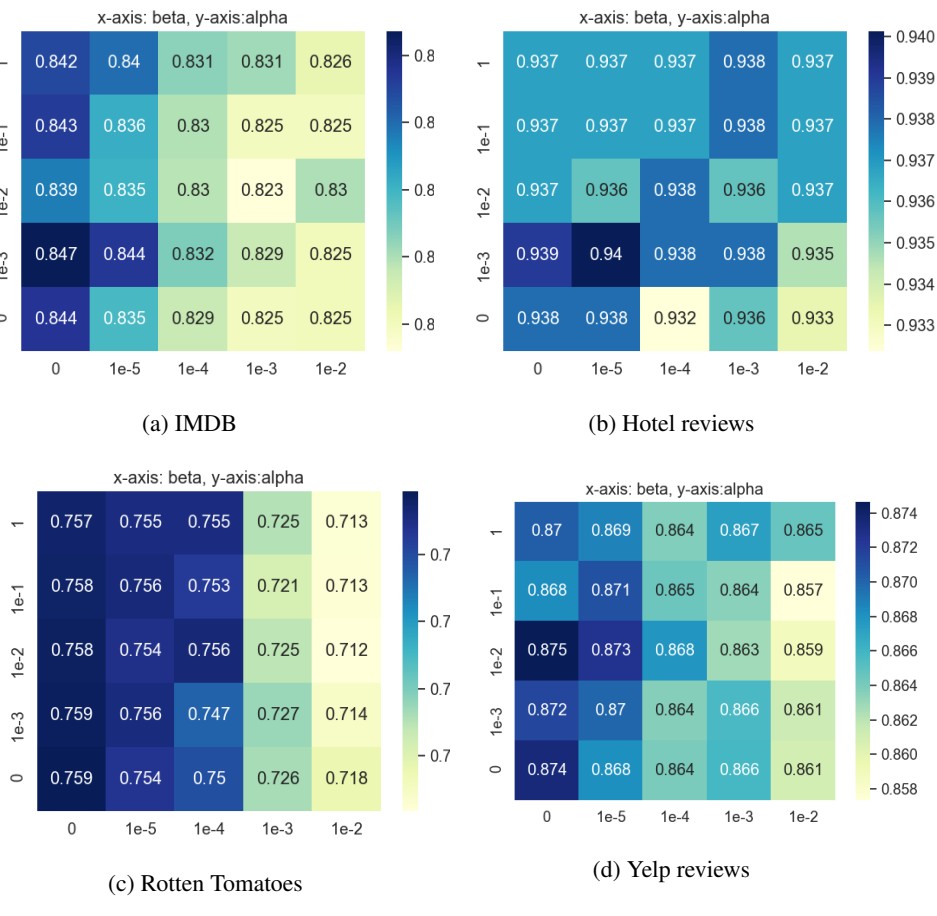

Figure 9: Effect of $\alpha, \beta$ on the model accuracy

| Sentences | Prototypes | Prototype score | Sentiment changes |
|---|---|---|---|
| [S1]The food here is very good. | Great menu, nice surroundings, good service and excellent food | 0.9080044 | |
| [S2]I did find the service a tad slow but of all the italian places in this area I would go here first. | My only complaint would be that it's a little small and it can get too crowded and busy, when that happens I'll go to the other Japanese restaurant right around the corner | 0.5125591 | |
| [S3] I had an italian hoggie that was very good and yes they do have exce llent meatballs | Appetizers Bbq shrimp sauce is great with bread and the Crawfish ceviche was flavorful and spicy | 0.8834794 | |

(a)

| Sentences | Prototypes | Prototype score | Sentiment changes |
|---|---|---|---|
| [S1]The sandwiches are giant, the atmosphere is good. | Great menu, nice surroundings, good service and excellent food | 0.9080044 | |
| [S2]I was there while the Steelers were on, so it was festive | With having a good experience previously, I decided to give Men's Wearhouse and Jos | 0.68709606 | |
| [S3] I agree they are a little dry | Not impressed whatsoever | 0.16809905 | |
| [S4] So get hot sauce, or ask for some mayo or ranch, etc. | The rice and beans are bland, but upon request they give a spicy sauce that tastes much like habenero sauce to me | 0.25679278 | |
| [S5] And it was quite warm in there | This place is great | 0.853906 | |

(b)

| Sentences | Prototypes | Prototype score | Sentiment changes |
|---|---|---|---|
| [S1] Boo. | Terrible service. | 0.04091281 | |
| [S2] What happened to this location. | Uh oh | 0.65373635 | |
| [S3] It used to be happy and fun. | Service was friendly and it was an all around great experience. | 0.89761466 | |
| [S4] It just feels depressing and nearly everyone there is just doing the bare minimum to get by. | I had just spent $30 there and that was totally wrong, that really put a sour taste in my mouth and i will not be a returning customer. | 0.10304821 | |
| [S5] So sad | Terrible service. | 0.04091281 | |

(c)

| Sentences | Prototypes | Prototype score | Sentiment changes |
|---|---|---|---|
| [S1] Awful everything. | Terrible service. | 0.04091281 | |
| [S2] From the bad service, to the diluted soup, to the lame-no chicken on the chicken salad, to the PRE-fabricated pizza. | The order was not made correctly, no cheese on the whopper with cheese, the fries were cold and soggy, and the chicken strips were most likely picked up from the floor. | 0.10424823 | |
| [S3] AVOID THIS PLACE | I won't be back. | 0.03598875 | |

(d)

Figure 10: The prototypes and sentiment scores used in Figure 4.

### 6.3.2 SURVEY QUESTIONS

Figures below show a few examples of the survey questions we used for the user evaluation study.

For the prototype selection, we created 10 questions each, for ProSeNet and ProtoryNet. Here we only show one example in Figure 11.

Figure 12 and Figure 13 show how we educated the subjects about how ProtoryNet or ProSeNet work.

For diagnosing the ProSeNet and ProtoryNet, we create 3 questions for each model. We show one example for each model in Figure 14 and Figure 15.

Select the sentence with the most similar sentimental semantics to the target sentence

**Target Sentence:**

It was delicious

A: They are great

B: Needless to say, we did not sign up for any of the plans and never returned

C: Great location but low on amenities

D: None of the above

Figure 11: Prototype selection question.

Now we present to you a model ProtoryNet that is based on the similarity between sentences and "prototypical" sentences to make a prediction. A prototype sentence is a sentence that is selected by the model to represent a group of sentences with similar meanings.

Let us a score between 0 and 1 to represent whether a sentiment is positive or negative. Larger values (closer to 1) indicate more positive sentiment and smaller values (close to 0) indicate more negative sentiment.

ProtoryNet first maps each sentence in a paragraph to the most similar prototypical sentence and then based on the trajectory of the sentiment in a paragraph, the model determines the overall sentiment of a paragraph

[Example] Here's a review of four sentences

[s1] food was delicious at this pace. [s2] But the service is super low and we waited for more than half an hour for our desert. [s3] When we asked the waitress about it, she was incredibly rude to us. [s4] We will not go back again.

The model finds the following prototypes for each sentence.

**Prototype for [s1]**: great food (*sentiment*: 0.9)

**Prototype for [s2]**: there is always a long wait for the food （*sentiment*: 0.2)

**Prototype for [s3]**: the service is very bad (*sentiment*: 0.05)

**Prototype for [s4]**: Overall, it was not worth it (*sentiment*:0.1)

We can visually observe how the sentiment changes. Sentiment drops from 0.9 to very low values.

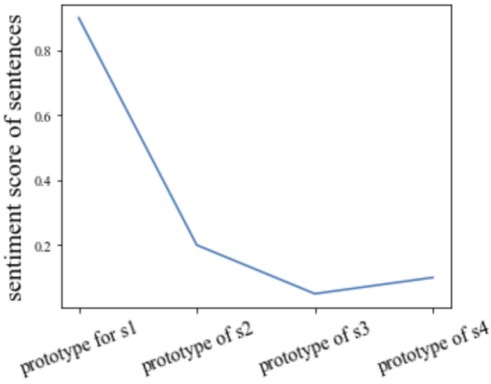

Figure 12: Education material for ProtoryNet

```
Input: excellent food . extremely clean . the staff is friendly and efficient .
       really good atmosphere .
Prediction: Positive(1.00)
Explanation:
(0.86)  EXCELLENT FOOD . NICE decor . a little pricey for the proportions . the
        salmon WAS AMAZING . ->  Positive(1.58)
(0.68)  GREAT FOOD SERVICE and views . hard to beat . this is our FAVORITE
        <other> restaurant in the valley . ->  Positive(1.23)
(0.50)  GREAT FOOD GREAT SERVICE . vietnamese <other> rolls unbelievable . GREAT
        food just what the neighborhood needed . ->  Positive(2.82)
```

Similarity between the input and each prototype
For example, 0.86 is the similarity between the
input and 1st prototype

Confidence of the prototype (how much a model
believes a prototype is positive or negative)

The total score is 0.86 * 1.58 + 0.68 * 1.23 + 0.5 *2.82 = 3.6052 >0
So the sentiment of the review is positive

Figure 13: Education material for ProSeNet

**Example 1**

**True Sentiment**: Negative.  **Model prediction**: Positive

**Review**: [s1] Don't go. [s2] I got more problems and sounds on my car after I spent $800 there. [s3] Unbelievable!

**Explanations**
[s1] -> prototype: Definitely won't be going back to this location (sentiment: 0.005)
[s2] -> prototype: I'd be willing to bet they get SO MANY complaint letters they just can't keep up(sentiment: 0.198)
[s3] -> Prototype: but eh (sentiment: 0.683)

Can you identify which prototype caused the incorrect prediction of the input?

A: prototype 1

B: prototype 2

C: prototype 3

D: I can't decide

Figure 14: Diagnosis question for ProtoryNet model

Example 1

**True Sentiment**: Negative.  **Model prediction**: Positive

**Input**: Don't go.  I got more problems and sounds on my car after I spent $800 there.  Unbelievable!
**Explanations**
(0.732 ) Prototype 1: Kuhn's automatically receive 1  for being open 24 hours. Beyond that, customer service has been great, the shelves are stocked well, prepared food and deli items are better for you than fast food with better ingredients at a better value! When I'm in there, this is my go-to shopping spot.        Positive (0.784)

(0.718)  Prototype 2: This Starbucks is teeny-tiny! Seating inside is VERY limited.  This is a Starbucks to grab and go and continue your shopping at the Waterfront. Baristas are friendly and fast.   Negative(0.445)

(0.715) Prototype 3: Exceeded my expectations! I had the fried chicken. It was tender and not greasy. The yams were tasty. Sweet but not overbearing. I will definitely visit again when I'm in the area!    Negative (0.751)

Can you identify which prototype caused the incorrect prediction of the input?

prototype 1

prototype 2

prototype 3

I can't decide

Figure 15: Diagnosis question for ProSeNet model

