# OpenReview forum: "Interpretable Sequence Classification Via Prototype Trajectory"
_ICLR.cc/2021/Conference — Reject_

### Official Review · AnonReviewer2 · 2020-10-24
**Incremental improvement on interesting concept with a thorough eval**

**Rating:** 4
**Confidence:** 5

**Review:**

This paper addresses the problem of explainable AI by trying to build models which are inherently interpretable (as opposed to post-hoc methods that interpret black-box models after being trained).

In particular, they focus on a recent paper on prototype-based networks, and a model called prosenet. They introduce an extension, protorynet, which fares better on longer documents by applying prototypes to each sentence (rather than the whole document), leading to a "prototype trajectory", showing how the prediction evolves over the course of the sentence.

They show improvements in prediction accuracy across 5 datasets, analyse the hyperparameters and conduct a human study.

Strengths:
I really like the concept of prototype networks, as it is simple and I could see it being accessible to non-technical users. The authors are to commended for undertaking a proper evaluation, including human experiments, which are often painful to conduct but quite helpful.

The proposed extension is straightforward and easy to understand. As someone unfamiliar with the original paper from Ming et al, I found it easy to get up to speed. The authors are also good at plainly describing the details of their approach, and running the requisite ablation studies (e.g. between sigmoid/exponential).

Weaknesses:
1. The paper's prior work section is incomplete, and makes factually incorrect statements about the state of the field. In particular, the authors focus on attention-based techniques, ignoring the considerable amount of work on other approaches, and incorrectly saying that other approaches "often turn out to be gibberish". Attention is a particularly strange subset of interpretations to focus on, given that the community has recently started to argue about whether attention is an explanation at all [7]

In general, I would suggest the authors look at papers from the “Interpretability and Analysis of Models for NLP” track at ACL and the blackboxNLP workshop at ACL. For concrete starting points, I'll restrict myself to ICLR papers, dating back to 2017 [2-5]. General attribution methods, such as integrated gradients [6] are also pertinent.

I don't think the paper can be published without a reasonable related work section, hence the "clear reject" score. If this were fixed, I'd upgrade to a "weak reject", i.e. 5.

2. I am concerned that this model may be outperformed by simple, bag of words approaches. For IMDB, the original paper [1] has a variety of results hitting 88% accuracy, while the results reported for protorynet are 85%. If a bag of words model outperforms protorynet, that makes it a less desirable model. It is possible that preprocessing differences account for this, though.

Can the authors give some clarity on this, for IMDB as well as the other datasets? Ideally, there would be an additional column in the results table providing results for a simple, non-neural, baseline.

3. I worry that a lot of the added accuracy does not help the model's accuracy. In particular, the ablation charts in Figures 5 and 8 show that when alpha=beta=0, the model's accuracy is within ~.2% of the best accuracy, so do we need those additional loss terms? I also suspect that averaging the outputs, rather than feeding them through an RNN would be comparably accurate, and simpler.

4. While I appreciate the human studies, the confidence bars are quite wide, so that most of the findings are not statistically significant.

5. For the model diagnosis human experiment, the users were only shown three examples. How were these examples chosen? That is not very many, so I worry that either those examples were chosen to be ones where protorynet was better. Ideally they would be chosen randomly, subject to some reasonable criteria.

Nitpicks:
- Distillbert is an odd SOTA to choose, as it is designed to have fewer parameters. Something like RoBERTA would likely have stronger results, and be more representative of SOTA.

[1] https://ai.stanford.edu/~amaas/papers/wvSent_acl2011.pdf
[2] https://arxiv.org/abs/1911.06194
[3] https://arxiv.org/abs/1812.04801
[4] https://arxiv.org/abs/1801.05453
[5] https://arxiv.org/abs/1801.05453
[6] https://arxiv.org/abs/1703.01365
[7] https://arxiv.org/abs/1902.10186

---

> ### Author Response · Authors · 2020-11-20
> **Response to Reviewer 2**
>
> Thank you for your positive feedback on our idea and concept! We apologize for not presenting enough material and let us try if we can address your concerns in the response.
> ***
> **Q1**:  lacks related work
>
> **A1**:  We apologize for missing the important related work. We want to clarify that "often turn out to be gibberish" refers to attention methods. We agree with you attention methods have been under heated debate and are particularly controversial. But from the feedback we have got so far, we realize many researchers would still immediately think of attention methods when it comes to interpretability of RNN. So we feel it is necessary to include it, even just to make a distinction. We will lightly touch attention methods in the updated version and cite the recent paper [4]. In addition, thank you very much for pointing us to the papers.  We have added one paragraph in the related work section discussions the papers listed in the review.  Also, we do want to make a distinction here, that the papers listed in your review are posthoc explanation methods while ProtoryNet itself is interpretable and does not rely on any external explainers.
> ***
> **Q2**:  Concern about the predictive performance and adding an  interpretable baseline
>
> **A2**:  We understand your concern about the predictive performance. We are working on re-running all experiments, this time allowing the BERT sentence embedder to be fine tuned (per Reviewer 3’s suggestion).
> We have added bag-of-words as an interpretable, non-neural baseline. (Note that in the paper you point us to, they used a dataset with 50k instances while our model was applied to the original version of the data with only 25k instances). We have run bag-of-words on our datasets for a meaningful comparison. On average ProtoryNet is better in predictive performance. Also, bag-of-words' explanations are on word-level, which might be too fine-grained. ProtoryNet provides sentence level interpretations and also track how sentiment changes.
> ***
> **Q3.1**:  the purpose of the diversity and prototypicality terms
>
> **A3.1**:  We apologize for not explaining the purpose of the two terms clearly in the paper. These two terms are NOT designed for improving accuracy, but for the purpose of improving interpretability. The fact that it doesn’t hurt the predictive performance can be explained by the recent research on “Rashomon Set”[1], that there exist many models with very similar performance, so one can add customized constraints to the model to achieve additional benefits, such as interpretability. Here, to achieve good explanations, we desire prototypes that are different from each other to avoid redundancy, thus the diversity term. We also want each input sentence to be mapped to a prototype that is similar enough to make the explanation convincing (if "I love the food" is mapped to the prototype "terrible service", users wouldn't believe this model), thus the prototypicality term. In fact, similar terms have been introduced in other prototype based DNN models [2,3]. We have added more explanations of the two terms in the paper.
> ***
> **Q3.2**: Averaging the outputs
>
> **A3.2**: Thank you for proposing the easy but interesting baseline. We have been running experiments. The results we have collected so far show that this baseline is worse than ProtoryNet. We will update the paper once we have complete results for all datasets.
> ***
> **Q4**: Statistically insignificant results
>
> **A4**:  The results were collected from 58 subjects in the original paper. In the last a couple of months, our survey continued to collect data and now we have a total of 111 subjects who participated in the survey. We will update the paper with the new results and report the p-value for comparing the two models.
> ***
> **Q5**:  Survey design
>
> **A5**: Each user was shown three examples, but the three examples were randomly drawn from a pool of 20 examples where both ProtoryNet and ProSeNet misclassified. The 20 examples were selected from the common set of misclassification and having less than or equal to 5 sentences.
> ***
> [1] Rudin, Cynthia. "Stop explaining black box machine learning models for high stakes decisions and use interpretable models instead." Nature Machine Intelligence 1.5 (2019): 206-215.
> [2] Chen, Chaofan, et al. "This looks like that: deep learning for interpretable image recognition." Advances in neural information processing systems. 2019.
> [3] Ming, Yao, et al. "Interpretable and steerable sequence learning via prototypes." Proceedings of the 25th ACM SIGKDD International Conference on Knowledge Discovery & Data Mining. 2019.
> [4] Jain, Sarthak, and Byron C. Wallace. "Attention is not explanation." arXiv preprint arXiv:1902.10186 (2019).
> ***
> **Please let us know if there's anything unclear or need more clarification**
> Thank you again for your comments!

---

> > ### Author Response · Authors · 2020-11-20
> > **new results for Q4**
> >
> > Dear reviewer,
> >
> > We have updated the human evaluation to respond to **Q4**. Our new evaluation was done on 111 subjects and we reported p-value for comparing the two models for technical users and non-technical users.

---

> > > ### Comment · AnonReviewer2 · 2020-11-20
> > > **Quick turnaround! More questions**
> > >
> > > Kudos for running new results so quickly, and getting them into the paper.
> > >
> > > Regarding your general comment above - I absolutely agree that this is an important research area, with interesting ideas, that should be pursued, and papers shouldn't be arbitrarily rejected.
> > >
> > > That being said, I still have concerns.
> > >
> > > 1. Performance
> > > Most importantly, it does not look like the new model beats bag of words. I computed the average performances by hand, and got 86.68 for BOW and 86.28 for ProtoryNet (it is possible I messed up this calculation). While BOW gives word-level explanations, you could easily process them to do sentence level explanations (by summing up the words in a sentence), or trajectories.
> > >
> > > I'm also not entirely convinced the provided BOW results are as strong as possible. SOTA. What would convince me is citing a paper, e.g. fasttext [1], that provides comparable results on those datasets.
> > >
> > > This is a valuable line of work, but meaningfully beating BOW is an important baseline.
> > >
> > > As an aside (and this doesn't impact your score), I suspect that incorporating a BERT-type architecture into ProtoryNet would be the secret sauce to getting better accuracy.
> > >
> > > 2. Hyperparameter selection
> > > How are alpha and beta chosen? I'm concerned the authors did trial and error to produce the best-looking interpretations. It's important for there to be a systematic way to tune hyperparameters on new datasets for users not intimately familiar with ProtoryNet. (This wasn't a concern before, as I assumed alpha and beta were chosen to optimize validation set accuracy).
> > >
> > >
> > > [1] https://arxiv.org/pdf/1607.01759.pdf

---

> > > > ### Author Response · Authors · 2020-11-20
> > > > **About comparing BoW and ProtoryNet**
> > > >
> > > > Thanks for the quick reply.
> > > >
> > > > We appreciate you carefully comparing BoW and ProtoryNet. While we doubt whether it is meaningful to average the accuracy across different datasets, we get that your concern is ProtoryNet does not clearly beat BoW by a margin. To this point here are our thoughts.
> > > > ***
> > > > 1) About the interpretability of BoW, while you can aggregate words in a sentence, you get the contribution of each sentence (suppose using a linear model so there are word-level weights). There is not a trajectory, but more like an attribution method where you know much each sentence contributes to the outcome. But one does not directly see how sentiment changes across the document. The sign of the weights are not indicative of the sentiment of sentences since everything is relative to the intercept, whose "sentiment" is unknown. We have not seen any literature that links BoW weights aggregated from words to sentences to the sentiment of sentences and we feel it is unnatural to do that.
> > > > ***
> > > > 2) We checked the fasttext paper. This paper doesn't talked about BOW and it talks bout a CNN model for text classification, which is a black box. It did cite paper [1] for BoW so we checked paper [1] for the experimental details. We found that the datasets they used are different from ours. For example, the Yelp review they use came from  Yelp Dataset Challenge in 2015. This dataset contains 1,569,264 samples that have review texts. Our Yelp data has 555,00 instances. Their Amazon review dataset came from the Stanford Network AnalysisProject (SNAP), which spans 18 years with 34,686,770 reviews while our Amazon review data was found on Kaggle. The remaining three other datasets we used do not overlap with theirs so we can not directly compare with their results.
> > > > ***
> > > > 3) We didn't do trial and error to produce good looking results. We set the parameters to fixed values as we said in the paper and explained in the response, which means they are the same for all datasets, to produce the results in the original paper. We didn't tune the parameters because we want to make it easy for users who do not know ProtoryNet, to make sure that even if you go with the default parameters, you can still achieve good results. We did show that, ProtoryNet is better than ProSeNet and (at least comparable) to BoW. Knowing that accuracy is the primary focus of reviewers,  we are tuning the parameters now. We are using a validation set to tune the performance. Now $\alpha$, and $\beta$ are randomly chosen from [ 0, 1] and K is chosen from drawn from [20, 500].  For reproducibility, all codes and data will be made public upon publication.
> > > > ***
> > > > **Please let us know if you have more suggestions and concerns.** We'd love to discuss more about the paper with you!

---

> > > > > ### Comment · AnonReviewer2 · 2020-11-21
> > > > > **Response**
> > > > >
> > > > > 1. I think it'd be entirely natural to process BoW, and a fairly obvious baseline. Simple, stupid baselines are very valuable.
> > > > >
> > > > > 2. Fasttext itself is actually a linear model, and would be a great benchmark, given that it's ~5 years old, well established (~2.4k citations) and has a widely used (and easy to use) github implementation with 22k stars. For prediction results, it's ideal to have results from other papers. Failing that, using a standard software implementation like fasttext, or vowpal wabbit, that has been properly tuned is a good substitute.
> > > > >
> > > > > 3. I do not understand - did you pick the numbers beta=.1 and alpha=1e-4 out of thin air without looking at the data at all, using some type of gut instinct? If I take any dataset in existence, those numbers will produce optimal results? But now they are being chosen to optimize predictive accuracy, which is what I thought they were originally being done for? And if that's the case, then my original concern still stands. I'm not trying to be difficult (or rude), but this should be a very simple, one sentence answer, and if it's not then that's a problem.
> > > > >
> > > > > Releasing the code is a great thing.

---

> > > > > > ### Author Response · Authors · 2020-11-21
> > > > > > **Response to Comment 3**
> > > > > >
> > > > > > Regarding comment 3, yes we used beta = .1 and alpha = 1e-4 for all datasets to produce the results in the original paper. This is the default setting we'd recommend like any algorithm would have some default parameters. Of course, they don't produce optimal results but will provide some reasonable results.
> > > > > >
> > > > > > Yes now we are tuning the parameters on a validation set. Can you please explain why it is a concern? We are rather confused at this point whether we are expected to tune the parameters or not tune them.
> > > > > > " ... But now they are being chosen to optimize predictive accuracy, which is what I thought they were originally being done for?" -> Yes we are doing parameter tuning using a validation set to improve the performance. " And if that's the case, then my original concern still stands" -> Could you please clarify？We don't understand what is your concern. If you are referring to "I'm concerned the authors did trial and error to produce the best-looking interpretations. ", then isn't this standard parameter tuning on a validation set? Why is it a concern?

---

> > > > > > > ### Comment · AnonReviewer2 · 2020-11-21
> > > > > > > **Clarification**
> > > > > > >
> > > > > > > Perhaps it'd be useful to clarify my expectation for hyperparameter selection. When a new algorithm has hyperparameters to tune, ideally the method's creators should provide the following 2 characteristics:
> > > > > > >
> > > > > > > 1. Intuitive explanation of what purpose the hyperparameter serves (e.g. more/less regularization)
> > > > > > >
> > > > > > > 2. An algorithm to select that hyperparameter that is linked to it's intuitive explanation
> > > > > > >
> > > > > > > 3. (Less important) validation that the hyperparameter is useful - i.e. we couldn't just set it to zero and forget about it.
> > > > > > >
> > > > > > > Originally, (paraphrasing) the authors provided the below responses
> > > > > > > 1. The hyperparameter helps make the models interpretations "better", in some sense
> > > > > > > 2. Unclear initially, but apparently set at fixed, arbitrary values
> > > > > > > 3. Unclear
> > > > > > >
> > > > > > > In this case, The responses to 2 and 3 are not great.
> > > > > > >
> > > > > > > Now, the authors are saying the below:
> > > > > > > 1. The hyperparameter is used to increase predictive performance
> > > > > > > 2. Standard optimization on validation set
> > > > > > > 3. Unclear, but it seems that the hyperparameter is not useful, as I originally noted that alpha=beta=0 produces pretty strong predictive results
> > > > > > >
> > > > > > > In this case, the response to 3 is not great.
> > > > > > >
> > > > > > > This is really important for your model! Based on the changing messages, it sounds like the authors themselves are not terribly sure. This is a problem for accepting the paper in its current form (though I suspect that with a bit more work the authors could figure this out, and resubmit to a later conference).

---

> > > > > > > > ### Author Response · Authors · 2020-11-21
> > > > > > > > **Response**
> > > > > > > >
> > > > > > > > Thank you for your clarification. I think the expectation is much clear now. We are glad that we are on the same page that we addressed point 1 and 2.
> > > > > > > > ***
> > > > > > > > But let us clarify point 1 before we address point 3. The purpose of the hyperparameters is to increase the predictive performance and/or interpretability. This part is also related to what we want to say about point 3.
> > > > > > > >
> > > > > > > > Now let us try if we can address 3, *validation that the parameters are important, i.e., why not just setting them to 0*.
> > > > > > > >
> > > > > > > > There are three parameters in this model,  $K$, $\alpha$, $\beta$. $K$ mainly has an effect on the predictive performance and as Figure 3 shows, larger $K$ has better performance. So the question is more specifically, *why not just setting $\alpha, \beta$ to 0?*
> > > > > > > > The short answer is, **because they help the interpretability while slightly improving the accuracy**.
> > > > > > > >
> > > > > > > > For $\alpha$ and $\beta$, this question is directly related to the purpose of the diversity term and prototypicality term, which we have added discussions in the paper.  Similar terms have been studied in [1][2]. They are designed to make sure prototypes are dissimilar enough from each other to avoid redundancy, and each input sentence is close enough to a prototype such that the explanation is convincing. With the literature [1][2] and our explanations in the paper, readers can have an intuitive understanding of the purpose of the two terms.
> > > > > > > >
> > > > > > > > To answer your question "why not setting them to 0", we are running experiments now setting $\alpha,\beta$ to 0 and then compute the robustness, a measure inspired from your earlier review. We plan to examine the 1) distances between input sentences and prototypes they are mapped to (for validating the prototypicality term) and 2) the distances between prototypes (for validating the diversity term), with and without the parameters set to 0.  If the two terms are effective, then we should expect the distances in 1) increase and the distances in 2) decrease for $\alpha,\beta = 0$.. We think this will answer your question of "why not setting them to 0".
> > > > > > > >
> > > > > > > > All authors are running experiments now to try to get the results before the deadline. **We are wondering whether you believe these new results are sufficient to address your comment 3. If you have other suggestions, please let us know.**
> > > > > > > >
> > > > > > > > Thank you!
> > > > > > > > ***
> > > > > > > >
> > > > > > > > [1] Chen, Chaofan, et al. "This looks like that: deep learning for interpretable image recognition." Advances in neural information processing systems. 2019. [2] Ming, Yao, et al. "Interpretable and steerable sequence learning via prototypes." Proceedings of the 25th ACM SIGKDD International Conference on Knowledge Discovery & Data Mining. 2019.

---

> > > > > > > > > ### Author Response · Authors · 2020-11-24
> > > > > > > > > **Addressed comment 3 with more results**
> > > > > > > > >
> > > > > > > > > Dear reviewer,
> > > > > > > > >
> > > > > > > > > We have added more results as promised to address your comment 3 -  "Unclear, but it seems that the hyperparameter is not useful".
> > > > > > > > >
> > > > > > > > > Let us reiterate that the diversity and prototypicality terms are designed for **interpretability** while not hurting accuracy due to the **Rashomon Set** theory. Our results in Section 4.3 explains why not set them to 0. Let us summarize it here also.
> > > > > > > > >
> > > > > > > > > 1) if setting $\alpha, \beta$ to 0, then sentences will be mapped to prototypes that are less similar enough to each other (shown by larger distances in Figure 5a).
> > > > > > > > > 2) if setting $\alpha, \beta$ to 0, then prototypes will be similar to each other (shown by smaller minimum distances in Figure 5b)
> > > > > > > > >
> > > > > > > > > We hope the new results addressed your concern and make it more clear the effect of $\alpha, \beta$ and the two loss terms. If there are still things unclear to you, please let us know.
> > > > > > > > >
> > > > > > > > > Thank you!

---

> > > > > > ### Author Response · Authors · 2020-11-21
> > > > > > **Regarding comment 1 & 2**
> > > > > >
> > > > > > Fasttext relies on Continuous Bag of Words, which learns an embedding of words. Then models using these representations as features are arguably interpretable, whether it's linear model or not,  because now the features do not have human understandable meanings. In the Fasttext paper, the authors choose h (the dimension of the text representation) to be 10, so for any user, technical or non-technical, it very difficult to trace back to the sentiment of each sentence, or at least, not that straightforward as ProtoryNet. This is why for structured data, some recent works start to use prototype-based explanations since it hide the complicated representations and calculations from users while still making the decision-making process easily understandable.
> > > > > >
> > > > > > We hope this can help clarify the confusion.
> > > > > >
> > > > > > Thank you!

---

> > > > > > > ### Comment · AnonReviewer2 · 2020-11-21
> > > > > > > **Response**
> > > > > > >
> > > > > > > Fasttext is a linear model with a low rank factorization constraint on the matrix. The  equation on page 2 of the fasttext paper shows that. To interpret it, you can get a single coefficient for each word (or K - 1 coefficients, for K class classification).
> > > > > > >
> > > > > > > There are plenty of papers doing linear, BoW models for text models. Feel free to pick any of them to compare against. The important part is that you do, in fact, compare against a, at least somewhat, SOTA BoW model.

---

> > > > > > > > ### Author Response · Authors · 2020-11-21
> > > > > > > > **more clarification needed**
> > > > > > > >
> > > > > > > > We have followed the "Bag-of-words and its TFIDF" in Section 3.1 in paper [1] to get the features and then apply Logistic Regression because we want our model to be interpretable. (The code will be posted for any validation of the results). We believe this is a quite reasonable SOTA BoW. **Do you think this is sufficient or you have other suggestions?** Please let us know so we have time to run more experiments. While we still believe Fasttext is not easily interpretable as ProtoryNet and thus a not good interpretable baseline, but if this is your major concern, please feel free to let us know.
> > > > > > > >
> > > > > > > > Thank you!
> > > > > > > > ***
> > > > > > > > [1] Zhang, Xiang, Junbo Zhao, and Yann LeCun. "Character-level convolutional networks for text classification." Advances in neural information processing systems. 2015.

---

> > ### Author Response · Authors · 2020-11-20
> > **new results for Q3.2**
> >
> > We have included the comparison with averaging the outputs in the paper.

---

> ### Comment · AnonReviewer2 · 2020-11-24
> **Wrapping up responses**
>
> There has been a lot of back and forth, so for simplicity (and the AC's benefit), I'm summarizing here.
>
> I'm leaving my review score at 4, and increasing my confidence score to 5.
>
> While the authors fixed my concerns about related work, interacting with them has made me more concerned about other aspects of their paper. To summarize/reiterate,
>
> 1. I don't trust the experimental results.
> a) The authors implemented a basic BoW model, which surprisingly does nearly the same as their LSTM model on datasets (especially IMDB) where LSTMs are known to perform better. This makes their initial claims about ProtoryNet outperforming LSTMs far more questionable.
> b) The authors have declined to cite baseline results from other papers, indicating they use different dataset processing. They have also declined to test against standard, widely used, linear models, such as fasttext.
> c) A point in my initial review that was ignored: their SOTA column uses a model that is not SOTA, and substantially understates how well existing, black box models do. E.g. IMDB lists a 92.3% accuracy, when the best method hits 96.2% [1]
>
> 2. Even if I did trust the experiments, they aren't very impressive. ProtoryNet performs basically the same as a run of the mill BoW model. Beating a bag of words model is not a terribly high bar, and should be a prerequisite for acceptance.
>
> 3. Hyperparameter selection. Their method introduces two hyperparameters, alpha and beta, which the authors mysteriously set to 0.1 and 1e-4. When pressed, they cannot say how they chose those values, nor how well they would apply to new datasets. While the authors provide some general intuition, there is no data-driven qualitative or quantitative validation that those terms are needed.
>
> 4. Incremental - given the ideas contained in Ming et. al, this paper is a relatively straightforward modification of an existing architecture.
>
> 5. The example interpretations in Figure 10 are actually qualitatively fairly hard to understand, and it is not clear why many of the input sentences are mapped to the corresponding prototypes (For context, this reviewer has spent a great deal of time looking at explanations of NLP models).
>
> 6. While the authors have made some improvements, the paper is still not very well written, and often challenging to parse.
>
> [1] http://nlpprogress.com/english/sentiment_analysis.html

---

> > ### Author Response · Authors · 2020-11-24
> > **Wrapping-up responses from the authors 1**
> >
> > Dear reviewer,
> >
> > As the rebuttal phase coming to an end, we'd like to address your comments one last time since we interacted with you most in the past few days and spent most of the time addressing your comments, which, at this point, we think we have addressed them all, with all authors spending a lot of time running experiments to fully respect your feedback. However, we do not feel we receive the same respect for our efforts since the results we generated have been ignored in your review.
> > ***
> > 1. a) We have the BoW model we used for comparison and you questioned whether the BoW model was state-of-the-art (were you thinking the BoW is too bad?) After we described to you how we did it and cited the paper we followed, now you question whether LSTM was properly done, because  BoW can perform similarly (now BoW became too good?).
> > b) We could not cite the results in other papers because, exactly like you said, they are different datasets. We do not have enough time to re-run the experiments on the exact datasets in that paper as you required. We are sorry for that. If the rebuttal phase could be longer in the future, such requests would be more feasible.  We did NOT refuse to test against standard linear models. You said that any state-of-the-art BoW would do. We implemented BoW+linear from paper [1] which was no doubt a state-of-the-art BoW, but we still asked you whether you think it was sufficient or you insisted us using fasttext. We never got our reply. c)  Yes there might be better SOTA but the entire point of the paper is that it is an interpretable model. If we are already not beating DistillBERT, it already shows that there is a gap between our model and the black-box SOTA. This is the fact we already acknowledged.
> >
> > We have been offended multiple times because in one of your reviews, you were "concerned the authors did trial and error to produce the best-looking interpretations", on no basis at all. We feel deeply offended by accusing of violating the basic ethnicity of research and performing academic fraud. It is especially ironic and frustrating because you have criticized that there exists a gap between our model and SOTA and if we did or were to do any unethical tuning, we wouldn't be having this problem.
> > ***
> > 2. Our model is (at least) on par with BoW, beating BoW on three out of five datasets. Most importantly, it is a **different** type of explanation. BoW, even if aggregated to sentence level as you suggested, is an importance attribution method which is a different type of explanation. As I have explained in previous responses. You cannot directly link weights to sentiment since they have to be relative to the intercept, for which you have to come up with some "sentiment".
> > ***
> > 3. This part is what frustrates us most.  We feel you completely ignored our explanations and new results. Please see the new results in section 4.3, which are quantitative, and our review response on Nov 24. Many algorithms have default parameters, especially deep learning.  We have provided you the experiments. The message is that the performance is consistently good for different values of $\alpha$, $\beta$ so we set them to a fixed value to provide the results, which should be a very good thing because now users don't need to relentlessly tune the parameters while still getting good results.
> > So to answer your question again here, "how they chose those values": because they are consistently good, we just fixed them to values that we found to be consistently good (like you would set the learning rate to a default value to start with). If you were to tune them, you can choose them from [0,1], like the sensitivity analysis.
> > "how well they would apply to new datasets": our whole results in the original paper were generated from the fixed values, this already explained this question.
> > "While the authors provide some general intuition, there is no data-driven qualitative or quantitative validation that those terms are needed." is a completely false statement because the original sensitivity analysis in the paper already provides a quantitative validation of how they impact the accuracy and the new paragraph in section 4.3 shows how they impact the interpretability, both quantitatively. Both results were there before your wrapping-up review was posted
> > ***
> > 4. We respectively disagree. Our paper itself is the evidence so we will say no more.
> > ***
> > [1]  Xiang Zhang,  Junbo Zhao,  and YannLeCun. 2015. Character-level convolutional networksfor text classification. InNIPS.

---

> > > ### Author Response · Authors · 2020-11-24
> > > **continued**
> > >
> > > 5. "not clear why many of the input sentences are mapped to the corresponding prototypes": if this is a problem to you, then all prototype-based methods are probably not a good choice for you. Prototype-based methods, map instances to the most similar prototypes, based on their distances. This part is not something we create.  We understand that different users may have different definitions and requirements of interpretability (like some other users might ask, how did you get the weights for BoW?) that is why we should welcome various kinds of interpretable models, such that users have different choices. Some users, like you, may choose BoW and like the importance of attribution. Some users may want an explanation simply based on how the sentiment changes along a trajectory. Such diversity should be allowed in academia and science in general.
> > > ***
> > > 6. Thanks for the feedback. There is always improvement to make on a paper at any stage. We will continue working on the paper to make it better.
> > >
> > > We understand there is a lot of work for reviewers to do during the rebuttal phase. Please also understand that we as authors also did a lot of work, with the common goal to improve the paper and exchange ideas.  We will accept and respect each reviewer's decision, on the basis that there is no misclaim on things we didn't do but we actually did in the paper but missed by the reviewers.

---

> > > ### Comment · AnonReviewer2 · 2020-11-24
> > > **Last comment**
> > >
> > > In terms of the substance of my review, the authors response does not change my perspective.
> > >
> > > The authors, unfortunately, appear to have taken the approach of attacking me personally and generally being difficult. I do not appreciate this, and think it needs to be explicitly condemned.
> > >
> > > I have been more generous with my time than 99% of reviewers, engaging in a lengthy back and forth and reviewing multiple revisions to give the authors a chance to improve their score. But even if I hadn't, this is simply not acceptable behaviour. I get that it can be frustrating to receive bad reviews, but that's not an excuse.
> > >
> > > For the record, I can be convinced by authors, and in my reviews for a different ICLR paper I did flip from a reject to an accept based on author feedback. I simply did not find the author's feedback on this paper to be convincing. While I'm sorry that that is the case, it is the truth.
> > >
> > > I have given enough of my time to this, and have thoroughly evaluated the paper. I will not respond to further author responses.

---

> > > > ### Author Response · Authors · 2020-11-24
> > > > **Last response**
> > > >
> > > > All reviewers and authors work towards a common goal to improve the paper. We thank your time and effort in doing that.
> > > > Our paper can certainly be criticized and that is what the review process for, but not the ethics of authors.  We were being attacked in the first place about doing "trial-and-error to produce the best-looking interpretations". This is a serious and irresponsible comment for any reviewer to accuse authors, without any basis and it needs to be explicitly condemned.

---

### Official Review · AnonReviewer4 · 2020-10-26
**A framework for text data that classifies that explains through prototypes' results**

**Rating:** 7
**Confidence:** 4

**Review:**

This paper presents ProtoryNet, a framework for text data that classifies and explains the prototypes' results.   The key concept, that is the novelty of the work, is that this framework is based on sentence prototypes, called prototype trajectory in the paper. In particular, instead of working at the entity of the text level, the text is split into sentences and each sentence is analyzed by itself.  The structure of the framework is composed of a layer that encodes a text's sequences, followed by a prototype layer in which is computed the similarity among each sentence and the prototype trajectories. At this point, the sentences are represented in one-hot encoding: for each sentence, there is a bunch of zero and then a one for the most similar sentence prototype. This representation is used for the classification of the sentence, done using an LSTM structure. In this setting, the interpretation is given by exploiting the prototypes matched for the text under analysis.

The idea presented in the paper is really interesting: it allows for an interpretation based on prototypes that is simple and understandable while achieving acceptable prediction performance.

Pros
It is well written and easy to read. There is only a repetition of a “the” at the end of page 7.
The idea of the trajectory prototypes is quite interesting, especially because it can be employed for other kinds of sequence data.
The explanations obtained are compelling: the results obtained from the human evaluation showed excellent results, especially in the context of local explanations for non-experts.
 I appreciated the presentation of the framework using a picture due to the complicated structure.

Cons
A few words more about the diversity and the prototypically would have been useful in the objective functions. The reader is able to get a general idea, but maybe in the appendix, there could be something more to understand the claims fully.
The prediction performance of the model is fine, but not excellent. In my opinion, some experiments more about the prototype initialization would have been helpful in understanding if it can be a possible source of errors.
The evaluation of the explanations is only w.r.t. ProSeNet. However, there are other methods to compare with, such as LIME and SHAP, to name agnostic methods, but also attention-based explanations, such as LRP (Layerwise-Relevance-Propagation), NeuroX and Integrated Gradients (that are not cited even in the related work section).
What about the robustness of the explanations? Similar sentences are going to be represented by the same prototype or by prototypes that are similar? An analysis of the robustness of the explanations would be useful (fidelity, hit, and similar metrics could be used, but also methods such as ROAR-RemOve And Retrain or KAR-Keep and Retrain)
What about other classificators? The last part of the framework uses a LSTM to predict the sentiment of the sentence. However, due to the structure based on similarity, other classificators such as logistic regression or decision tree may be tested in the same fashion of shapelet-based classifiers. The explanation may also benefit from the use of such methods due to the additional interpretable information they provide.

[LIME] Ribeiro, M. T., Singh, S., & Guestrin, C. (2016, August). " Why should I trust you?" Explaining the predictions of any classifier. In Proceedings of the 22nd ACM SIGKDD international conference on knowledge discovery and data mining (pp. 1135-1144).
[SHAP] Lundberg, S. M., & Lee, S. I. (2017). A unified approach to interpreting model predictions. In Advances in neural information processing systems (pp. 4765-4774).
[LRP] Patil, A., Wadekar, A., Gupta, T., Vijan, R., & Kazi, F. (2019, July). Explainable LSTM Model for Anomaly Detection in HDFS Log File using Layerwise Relevance Propagation. In 2019 IEEE Bombay Section Signature Conference (IBSSC) (pp. 1-6). IEEE.
[LRP] Bach, S., Binder, A., Montavon, G., Klauschen, F., Müller, K. R., & Samek, W. (2015). On pixel-wise explanations for non-linear classifier decisions by layer-wise relevance propagation. PloS one, 10(7), e0130140.
[INTGRAD] Sundararajan, M., Taly, A., & Yan, Q. (2017). Axiomatic attribution for deep networks. arXiv preprint arXiv:1703.01365.
[SHAPELET] Lines, J., Davis, L. M., Hills, J., & Bagnall, A. (2012, August). A shapelet transform for time series classification. In Proceedings of the 18th ACM SIGKDD international conference on Knowledge discovery and data mining (pp. 289-297).

---

> ### Author Response · Authors · 2020-11-20
> **Response to Reviewer 4**
>
> Thank you very much for your valuable feedback! We are enthusiastic that you like our idea. Please let us do a little more explanation here to address your concern.
> ***
> **Q1**:  typo
>
> **A1**:  fixed. Thank you for your careful reading.
> ***
> **Q2**:  a few more words about the diversity and prototypicality
>
> **A2**:  We have added a paragraph in Section 3 to discuss the two terms in more detail. Please see the updated paper.
> ***
> **Q3**:  Predictive performance is fine but not excellent
>
> **A3**:  Yes we agree there is a gap compared to SOTA. We have tested not initializing the prototypes but the performance got slightly worse, so we can rule out it as an explanation. We are now tuning the hyper-parameters and the number of LSTM layers in order to improve the performance. (We set them to fixed values in the original submission)  We are also following Reviewer 3’s advice of fine-tuning the BERT embedding part (component a) together with the rest of the model (component bcd) during training to see if there’s an improvement. On the other hand, we want to get reviewers’ attention that the main advantage of ProtoryNet is the interpretability, for both technical users and non-technical users. So we went into great detail comparing ProtoryNet with the other very recent interpretable RNN model to show that our model is both more interpretable and accurate.  We have also added comparisons with two more baselines proposed by Reviewer 2. The results are shown in the updated paper. Please also see our comment to all reviewers at the top of the page.
> ***
> **Q4**:  Comparison with model agnostic methods such as LIME and SHAP, and attention-based explanations
>
> **A4**:   We have added the methods you pointed out in the paper in the related work. The main reason we did not compare with LIME and SHAP is that they are post-hoc explanation methods. Posthoc explainers explain a black-box predictive model. Here, ProtoryNet is a predictive model itself, so they are two types of models that are not directly comparable. Meanwhile, attention-based explanations are not comparable either, since recent research seems to be against using the attention weights as explanations [1]. Reviewer 2 also pointed out similar things that attention-based methods are not good comparison here. The most directly comparable baseline is ProSeNet, that is why we went into great detail to compare it with our model with both experimental evaluations and human evaluation. Please also see our discussion in response to all reviewers at the top of the page.
> ***
> **Q5**:  Robustness evaluation
>
> **A5**:  You made a very good point about the robustness evaluation! We will add a new metric in the paper, called epsilon-robustness, which measures the probability that two sentences are mapped to different prototypes if their distance is less than epsilon. We upload the results once the experiments are done. We have too many experiments running at the same time. Please allow us some time to get the new results. Thank you!
> ***
> **Q6**: Using other classifiers such as decision trees and logistic regression
>
> **A6**:  While each sentence is mapped to a prototype to be represented by a sparse vector, it is still sequence data with varying lengths, so decision tree or logistic regression models cannot be applied. But we have implemented the Reviewer 2's idea of directly averaging the sentiments of prototypes the input sentences are mapped to, and the performance is worse than using an LSTM (please see the results in the updated paper). If you have a specific idea of how to apply decision tree or logistic regression to replace LSTM, please let us know. Thank you!
> ***
> [1] Jain, Sarthak, and Byron C. Wallace. "Attention is not explanation." arXiv preprint arXiv:1902.10186 (2019).

---

### Official Review · AnonReviewer1 · 2020-10-28
**Good paper, minor concern**

**Rating:** 6
**Confidence:** 3

**Review:**

Summary: this paper presents an RNN sequence classifying model that generates a prototype for each sentence in a paragraph. The generated prototypes help explain the model's prediction. The method embeds each sentence, matches to prototypes, then runs through an LSTM before making a prediction. Experiments found improved accuracy compared to a previous model that generates only one prototype for a paragraph. A user evaluation also found improvement in interpretability.

Strengths:
-  The paper is well written and easy to understand.
- Generating prototypes is a promising direction for improving the interpretability of RNNs and other neural nets models. The idea of generating a prototype trajectory for sentences in a paragraph is interesting and novel to my knowledge.
 - The architecture and training methods are technically sound.
- The experiments show positive improvement in prediction accuracy.

Concern on user evaluation:
- There is some improvement but the error bars are large, it is not clear if the differences are statistically significant.
- With a prototype for each sentence, users will need to read more. So a more fine-grained explanation will increase cognitive load. User evaluation can potentially investigate this.

---

> ### Author Response · Authors · 2020-11-20
> **Response to Reviewer 1**
>
> Thank you very much for your positive comments. We are very glad you like our paper. Here we address your concerns on user evaluation.
> ***
> **Q1**:  Error bar being too large
>
> **A1**: The results were collected from 58 subjects in the original paper. In the last couple of months, our survey continued to collect data and now we have a total of 111 subjects who participated in the survey. We have updated the paper with the new results and report the p-value for comparing the two models. Results show that the difference is statistically significant.
> ***
> **Q2**:  Fine-grained explanation increases the cognitive load
>
> **A2**:  Thank you for bringing up this discussion. Here’s our thoughts on this topic. We think that users' comprehension of a new model follows a Bell curve, where as information is provided, they understand the model better. However, when the information saturates or overloads, their understandability is inhibited, just as you concerned. Without user study, it is unknown where the amount of information provided lies in the Bell curve. So that is why we designed the experiments in the paper, comparing ProtoryNet with ProSeNet, knowing that ProtoryNet presents more information than ProSeNet. The results already show that users feel ProtoryNet is easier to understand than ProSeNet. According to the comments we collected from the survey, users feel that prototypes provided at the document level are difficult to understand or map to the input since the semantics are more complicated; while in our model, prototypes at the sentence level are easier to make such links.
> In addition, we feel interpretability is quite subjective and has to be determined by the application and the specific user. Reviewer 2 proposed Bag-of-words, which is on the word level, ProSeNet is on the document level, and our model, ProtoryNet, sits right in the middle. In practice, it will be helpful to provide users different options so they can choose based on their specific task and application.
> ***
> **Please let us know if there's anything unclear or need more clarification.** Thank you again for your comments!

---

### Official Review · AnonReviewer3 · 2020-10-28

**Rating:** 5
**Confidence:** 3

**Review:**

The authors propose ProtoryNet,  a prototype-based model for paragraph classification that associates each sentence in the paragraph with a relevant prototypical sentence from the training data. The idea is interesting and the ability to decompose sentiment scores over each sentence + find prototypes for each helps to build user understanding of the model prediction. Thank you to the authors for the submission.

However, I have two concerns---baseline comparisons and model details---that prevent me from assigning a higher rating.

**Baseline comparisons**

1. Across 5 sentiment classification datasets, the authors find that ProtoryNet substantially underperforms a standard BERT model, in some cases obtaining more than double the error. This seems like a substantial price to pay the ability to associate each sentence with a prototype. The authors write that "Note, however, that DistilBERT was pre-trained on a massive corpus of text data... Hence, [it] should only be used for [a] sanity check". However, ProtoryNet also seems to build on top of standard pre-trained BERT embeddings, which have also been derived from a massive corpus of text data (and actually, the comparison should favor ProtoryNet, since DistilBERT is a smaller model than the standard BERT model). Could the authors elaborate on why they believe that this comparison is unfair? If it is unfair, the authors should perform a comparison that is as similar as possible to ProtoryNet but without the prototype parts, i.e., train a model on top of the same BERT embeddings and see how that performs.

2. Related to the above question, it is common to fine-tune BERT models on the dataset of interest. Was this done here for DistilBERT? What about for ProtoryNet? And if not, why not?

**Model details**

3. It was difficult to follow all of the model details; perhaps consider reorganizing and clarifying the writing. For example, it was unclear how the prototypes are actually chosen until late in the paper, whereas it should have been explained in S3.1. The notation in S3.1 has a few minor errors. For example, if the entire sentence is encoded as $\mathbb{R}^V$ then it seems like $V$ is not just the size of the vocabulary, but the size of the vocabulary to the power of the length of the sentence? Also, how were the hyperparameter values and coefficient values chosen? Is prototype projection also done at the end of training?

4. There are many modeling decisions that seem somewhat ad-hoc or non-standard. It seems like it might be possible to simplify the model significantly, or if not, it would be nice for the effects of these decisions to be better studied. For example: (a) mean-squared error is used even on binary classification problems; (b) the loss function is complicated by diversity and prototypicality terms, but the sensitivity analysis reveals that the accuracies are basically indistinguishable even when we completely remove those terms; (c) the sparsity transformation was approximated by a softmax that seems basically indistinguishable from a step function since $\gamma \geq 10^6$, so does it actually matter? (Note that ReLUs are also not differentiable.) (d) How useful is the LSTM at the end, if it generally goes over only ~4 sentences?

**Update**

Thank you to the authors for the revisions, and great to know that the experimental results have improved significantly. In the absence of an updated manuscript, it is difficult to update my score appropriately, so I will leave it as it currently is. However, I think the work is promising and that an updated manuscript that incorporates the new experimental results and more carefully teases apart the contributions of the different components (as the authors have started to do in this rebuttal period) would be impactful. Thank you to the authors again for all of their hard work.

---

> ### Author Response · Authors · 2020-11-20
> **Response to Reviewer 3**
>
> Thank you for your valuable feedback! Here we address your comments and clarify the paper a little more.
> ***
> **Q1**:  Performance comparison with DistilBERT
>
> **A1**:  you are right that DistilBERT outperforms ProtoryNet. We did not claim it was an “unfair” comparison and we acknowledged that ProtoryNet did not perform as well as the SOTA, paying the price for interpretability. However, ProtoryNet outperforms the latest prototype-based baseline ProSeNet. So the message we try to deliver is that if one desires such an understandable model, then ProtoryNet is a better choice than the other interpretable baseline. We understand that many researchers may still view predictive performance as a dominating factor when evaluating a new method, but we hope the interpretability of the model can be appreciated. Please also see our discussion of this issue in the response to all reviewers.
> ***
> **Q2**:  fine-tuning BERT
>
> **A2**:  We did fine-tune DistilBERT on each dataset of interest but we didn’t fine-tune BERT sentence embedding in ProtoryNet, using BERT as a service. You actually made a very good point! Thank you for that! We are re-running our experiments now, allowing the BERT sentence embedding to be trained with the rest of the model. But the training takes a long time so please allow us some time.
> ***
> **Q3**:  model details
>
> **A3**:  Thank you for the suggestion. Section 3.1 describes the architecture and **forward** functions from component a to d. We added explanations about the prototypes when we describe the prototype layer but we will leave the prototype initialization and projection to section 3.3 since it is part of the training process. Each sentence is coded as $\mathbb{R}^V$  because each sentence is a vector of size $V$, where each element in $V$ represents whether the corresponding word appears in the sentence. In the submitted paper we set $\alpha={0.1}$, $\beta=1e^{-4}$, and $K = 200$ to generate the results in Table 1. We are running new experiments tuning these parameters in order to improve the performance. The prototype projection is done every 10 epochs.
> ***
> **Q4**:  modeling decisions
>
> **A4**:  a) There exist many choices of loss functions such as MSE, cross-entropy, etc . MSE is one common choice. We do not see any particular reason for choosing or not choosing a specific loss function. So for the purpose of presenting the model, we just pick MSE as it can also be used for both classification and regression, which our framework can also be applied. If you feel there’s a specific loss function that needs to be tested, please let us know and we can run more experiments. b) the diversity and prototypicality terms are designed not for the purpose of improving accuracy, but for improving interpretability. The reason it doesn’t hurt the predictive performance can be explained by the recent research on “Rashomon Set”[1], that there exist many models with very similar performance, so one can add customized constraints to the model to achieve additional benefits, such as interpretability. Here, to achieve good explanations, we desire prototypes that are different from each other to avoid redundancy, thus the diversity term. We also want each input sentence to be mapped to a prototype that is similar enough to make the explanation convincing, thus the prototypicality term. In fact, similar terms have been introduced in other prototype based DNN models [2,3]. c) We apply a softmax not only to approximate a step function but most importantly, to select the most similar prototype (with the largest similarity). So it is necessary to apply a softmax instead of a step function. d) Our experiments actually show the performance of long paragraphs (>25 words, which are often over 4 sentences). We are running experiments now where we simply averaging the sentiments of the sentences. The results we have collected so far suggest that the performance is worse than using LSTM at the end to capture the temporal patterns.
> ***
> [1] Rudin, Cynthia. "Stop explaining black box machine learning models for high stakes decisions and use interpretable models instead." Nature Machine Intelligence 1.5 (2019): 206-215.
> [2] Chen, Chaofan, et al. "This looks like that: deep learning for interpretable image recognition." Advances in neural information processing systems. 2019.
> [3] Ming, Yao, et al. "Interpretable and steerable sequence learning via prototypes." Proceedings of the 25th ACM SIGKDD International Conference on Knowledge Discovery & Data Mining. 2019.
> ***
> **Please let us know if there's anything unclear or need more clarification**
> Thank you again for your comments!

---

> > ### Author Response · Authors · 2020-11-25
> > **Update on Q2**
> >
> > Here's the update on fine-tuning BERT.
> >
> > We are very pleased to find that your suggestion turn out to improve the performance significantly. Although still not being able to beat SOTA, but the performance now is much better than all baselines we compared with, including the new ones we added. Thank you very much for this valuable input!
> >
> > Due to the time constraint, we were able to get the results on one fold (instead of 5-fold in the paper) for 4 datasets. Here we report the accuracy on that one fold.
> >
> > | Datasets (one fold) | no fine-tuning (original paper) | fine-tuning  of BERT |
> > |:--------:|:-------------------------------:|:--------------------:|
> > |   IMDB   |              0.846              |         0.887        |
> > |  Amazon  |              0.879              |         0.925        |
> > |   Yelp   |              0.872              |         0.921        |
> > |   Hotel  |              0.950              |         0.965        |
> >
> > Thank you for your suggestion!

---

### Author Response · Authors · 2020-11-20
**Thank you to all reviewers!**

Dear Reviewers:

We thank all reviewers for their useful comments and great suggestions. Among the reviews, we noticed the two common concerns. Therefore, we would like to provide more clarification in this thread.
***
*First, about the predictive performance being worse than SOTA.*
ProtoryNet is an **inherently interpretable** neural network model. It is inherently interpretable, so it does not rely on post hoc explainer methods like DeepLift, Integrated Gradients, or SHAP to provide explanations. And its interpretability is intended for both technical users and non-technical users. However, this easily accessible interpretability comes at the price of accuracy. (Actually, we cannot find an **interpretable SOTA** that can beat **black-box SOTA**). Despite that, it is still more accurate than the interpretable baseline, ProSeNet, in addition to being more interpretable. We have also added bag-of-words, based on Reviewer 2's suggestion. ProtoryNet is still better on average.

Therefore, we hope reviewers can allow some time and opportunity for interpretable DNN models to develop and grow, before achieving the same or better predictive performance than SOTA, one day.

Meanwhile, now that we realize a lot of emphases has been placed on the predictive performance, we are running a series of experiments to further tune ProtoryNet, which includes allowing BERT to be fine-tuned instead of being used as a service, and tuning the hyper-parameters  $\alpha, \beta, K$ and the number of LSTM layers, which we set to fixed values in the original paper. Please allow us some time to finish the experiments.
***
*Second, about other alternative methods to achieve interpretability*
 We agree there are many explainer methods designed for DNN or RNN, such as Integrated Gradients and hierarchical explanation methods. But they are are post hoc and external **explainer** methods. ProtoryNet itself is a **predictive model** that is interpretable on its own. Inherent model interpretability is actually becoming more desirable since post hoc explanations may suffer from various issues according to recent research, such as low fidelity, inconsistency, and etc. We added recommended related work in the paper but we also want to make a distinction from these prior work.

In addition, bag-of-words, as recommended by reviewer 2, is also an interpretable model, but the explanation is on word-level which might be too fine-grained. However, we do not claim that ProtoryNet is more interpretable than bag-of-words since different domains, different applications, and different users may require different interpretability. It is hard to claim one form of model is better than another form of model, with totally different types of explanations. That is why we only compare with another prototype-based baseline, ProSeNet in the paper.
***
I hope the clarification here can bring reviewer's attention to our model's contributions and distinctions from prior work. If there's anything still unclear, I'm happy for any discussions.

Thank you!

---

> ### Comment · AnonReviewer2 · 2020-11-20
> **Updated paper?**
>
> The authors reference an updated paper, but the open review manuscript hasn't been updated.

---

> > ### Author Response · Authors · 2020-11-20
> > **Revision updated and waiting to add more experimental results**
> >
> > Sorry for the delay. We have been waiting to add more results in the paper. We have uploaded the version where we
> > ***
> > 1) enriched the related work
> > 2) added explanations of the prototypicality term and diversity term and
> > 3) added the results for bag-of-words.
> >
> > Changes are marked in red.
> > ***
> > We are waiting to get new results of ProtoryNet by tuning the parameters (which we all set to fixed values in the original submission).

---

> ### Author Response · Authors · 2020-11-25
> **Summary of improvements made during the rebuttal**
>
> Dear reviewers,
>
> Here we summarize the list of improvements we made to the paper, based on suggestions of reviewers.
>
> 1. enriched the related work and added papers recommended by reviewers
> 2. added explanations of the prototypicality term and diversity term
> 3. added the baseline of bag-of-words
> 4. added the baseline of averaging the sentiment of prototypes instead of using an LSTM
> 5. added in Section 4.3 the effect of $\alpha, \beta$ on interpretability
> 6. updated the human evaluation with more subjects (the experiments continued running since the submission of the paper so now we have 111 subjects in total) and added p-value to compare ProtoryNet and ProSeNet.
> 7. did hyper-parameter tuning of the model and updated the results (improved by ~1%)
> 7. Most importantly, we took Reviewer 3's advice to fine-tune BERT and we got a significant improvement of the results. However, due to the time constraint, we were only able to get the results on one fold for 4 datasets (instead of 5-fold in the paper) so we did not update the paper. Here we report the accuracy on that one fold.
> | Datasets (one fold: 80% training, 20% testing) | no fine-tuning (original paper) | fine-tuning  of BERT |
> |:--------:|:-------------------------------:|:--------------------:|
> |   IMDB   |              0.846              |         0.887        |
> |  Amazon  |              0.879              |         0.925        |
> |   Yelp   |              0.872              |         0.921        |
> |   Hotel  |              0.950              |         0.965        |
> We thank all reviewers for the valuable input!

---

### Decision · Program_Chairs · 2021-01-07
**Final Decision**

**Decision:**

Reject

**Comment:**

The authors introduce an RNN model, ProtoryNet, which uses trajectories of sentence protoypes to illuminate the semantics of text data.

Good points were brought up and addressed in discussion, which have improved the paper - including a helpful suggestion from Rev 3 to fine-tune BERT sentence embeddings in ProtoryNet, which led to significant performance gains.

Unfortunately the tone of discussion with one reviewer slipped below the respectful standards to which we aspire, but rest assured that only substantive points on the paper were considered.

Reviewers were split but in discussion converged to leaning against acceptance, allowing the authors to reflect on, and incorporate new results carefully in an updated manuscript.